EMBO
Molecular Medicine

# A novel P2X2-dependent purinergic mechanism of enteric gliosis in intestinal inflammation

Reiner Schneider[1] , Patrick Leven[1], Tim Glowka[1], Ivan Kuzmanov[1], Mariola Lysson[1],
Bianca Schneiker[1], Anna Miesen[1], Younis Baqi[2,3], Claudia Spanier[3], Iveta Grants[4], Elvio Mazzotta[4],
Egina Villalobos-Hernandez[4], Jörg C Kalff[1], Christa E Müller[3], Fedias L Christofi[4] & Sven Wehner[1,*]

## Abstract

Enteric glial cells (EGC) modulate motility, maintain gut homeostasis, and contribute to neuroinflammation in intestinal diseases and motility disorders. Damage induces a reactive glial phenotype known as "gliosis", but the molecular identity of the inducing mechanism and triggers of "enteric gliosis" are poorly understood. We tested the hypothesis that surgical trauma during intestinal surgery triggers ATP release that drives enteric gliosis and inflammation leading to impaired motility in postoperative ileus (POI). ATP activation of a p38-dependent MAPK pathway triggers cytokine release and a gliosis phenotype in murine (and human) EGCs. Receptor antagonism and genetic depletion studies revealed P2X2 as the relevant ATP receptor and pharmacological screenings identified ambroxol as a novel P2X2 antagonist. Ambroxol prevented ATP-induced enteric gliosis, inflammation, and protected against dysmotility, while abrogating enteric gliosis in human intestine exposed to surgical trauma. We identified a novel pathogenic P2X2-dependent pathway of ATP-induced enteric gliosis, inflammation and dysmotility in humans and mice. Interventions that block enteric glial P2X2 receptors during trauma may represent a novel therapy in treating POI and immune-driven intestinal motility disorders.

**Keywords** enteric nervous system; gut inflammation; motility disorders; postoperative ileus; purinergic signaling

**Subject Categories** Digestive System; Immunology; Neuroscience

## Introduction

Enteric glial cells (EGCs) are a unique population of cells in the enteric nervous system (Furness, 2012) playing a pivotal role in the maintenance of gut homeostasis (Sharkey, 2015). They shape the immune environment through interactions with resident immune cells and other cell types (Brierley & Linden, 2014; Yoo & Mazmanian, 2017). In line with this, EGCs secrete neuroprotective (Abdo *et al*, 2010) and immune-modulatory factors (Yoo & Mazmanian, 2017) and targeted ablation of glia (Rao *et al*, 2017) or inhibition (McClain *et al*, 2014) of glial signaling through connexin-43 hemichannel communication between glia can disrupt motility. However, the neuroinflammatory effect of glial ablation is still unclear, as in some cases a fatal bowel inflammation was documented (Bush *et al*, 1998; Cornet *et al*, 2001; Aubé *et al*, 2006) while in a recent study, utilizing a new genetic mouse model, no immune-modulatory effect was observed (Rao *et al*, 2017). In contrast to their immune-modulatory role, several *in vivo* and *in vitro* studies by us and others provide evidence that murine or human EGCs can turn into reactive glia in an immune-stimulated environment, e.g., under LPS presence (Rosenbaum *et al*, 2016; Liñán-Rico *et al*, 2016), after viral protein HIV-1 Tat (Esposito *et al*, 2017) or IL-1 stimulation upon which EGCs release inflammatory mediators like cytokines, nitric oxide or reactive oxygen species (Stoffels *et al*, 2014; Brown *et al*, 2016; Liñán-Rico *et al*, 2016; Rosenbaum *et al*, 2016). EGCs were also shown to interact with bacteria, and they can discriminate between beneficial and harmful bacteria (Turco *et al*, 2014).

Immune responses are often a consequence of tissue damage which leads to the release of intracellular molecules that act as danger-associated molecular patterns (DAMP) and trigger innate immune processes (Yoo & Mazmanian, 2017). One prominent DAMP is ATP that is produced and utilized by all cell types (Idzko *et al*, 2014). In the healthy gut, ATP is involved in intestinal homeostasis, gastrointestinal motility, blood flow and synaptic transmission (Christofi, 2008). However, increased extracellular ATP concentrations resulting from tissue damage and trauma, excessive mechanical stimulation, shear stress in diseased blood vessels, cancer, inflammatory cells or a variety of acute or chronic diseases represent a pathogenic pro-inflammatory mechanism contributing to symptomatology (Idzko *et al*, 2014; Di Virgilio *et al*, 2018).

ATP signaling is complex and is mediated by purinergic receptors to which ATP either binds directly or as an enzymatically metabolized form, e.g., ADP or adenosine (Galligan, 2008). Purinergic

1  Department of Surgery, University of Bonn, Bonn, Germany
2  Faculty of Science, Department of Chemistry, Sultan Qaboos University, Muscat, Oman
3  Pharmaceutical Institute, Pharmaceutical & Medical Chemistry, University of Bonn, Bonn, Germany
4  Department of Anesthesiology, Wexner Medical Center, The Ohio State University, Columbus, OH, USA
   *Corresponding author. Tel: +49 228 287 11007; E-mail: sven.wehner@ukbonn.de

receptors are classified broadly into ionotropic P2X, metabotropic P2Y and P1 receptor families. ATP, or other nucleotides can variably activate P2X and P2Y while adenosine activates metabotropic P1 receptors (Galligan, 2008). Recent studies demonstrated the expression of purinergic receptors on EGCs and their role in the regulation of gastrointestinal motility (McClain *et al*, 2014), neuron-to-glia communication (Gulbransen & Sharkey, 2009) and neuronal survival (Brown *et al*, 2016). We have identified P1, P2X and P2Y purinergic receptors in primary human EGCs in primary culture networks and the molecular identity of the reactive hEGC phenotype was revealed by LPS induction (Liñán-Rico *et al*, 2016). Recent progress in the field suggests that EGC may represent "a new frontier in neurogastroenterology and motility" (Ochoa-Cortes *et al*, 2016).

Overall, EGCs modulate motility, maintain gut homeostasis, and contribute to neuroinflammation in intestinal diseases and motility disorders (Gulbransen & Christofi, 2018). The latter includes postoperative gastrointestinal dysfunction and postoperative ileus (POI), a common clinical complication observed upon abdominal surgery that is characterized by a transient impairment of gastrointestinal (GI) function after surgery. POI is associated with increased morbidity in patients, and despite implementation of enhanced recovery protocols for elective colorectal surgery (Hedrick *et al*, 2018), no good treatment option exists. POI remains a huge health care problem costing billions of dollars in extended hospitalizations (Iyer *et al*, 2009). POI is well known to originate from postoperative neuronal dysregulation and is based on an inflammation of the muscularis externa (ME) (Wehner *et al*, 2007). Recently, we demonstrated that this postoperative inflammation involves EGC reactivity (Stoffels *et al*, 2014), but the molecular identity of the induction and trigger mechanisms of EGC activation are not fully understood.

Herein, we tested the hypothesis that surgical manipulation and trauma triggers ATP release that drives enteric gliosis and intestinal inflammation leading to impairment of motility in POI. We accessed the relevance of reactive EGC in human bowel specimens and the well characterized mouse model of acute posttraumatic bowel inflammation resulting in POI. By transferring the discovered mechanistic insights to a clinically relevant treatment option of selective purinergic receptor antagonism with ambroxol, a newly identified P2X2 antagonist "drug", we confirmed the potential therapeutic importance of ATP-activated EGCs for inflammation-induced POI that may be relevant to other motility disorders.

## Results

Enteric glial cells respond to injury and inflammation and contribute to damage and regenerative processes (Grubišić & Gulbransen, 2017). Our investigation uncovered a purinergic pathway in reactive murine and human EGCs involved in the response to surgical trauma and inflammation.

### ATP induction of a reactive EGC phenotype is dependent on a p38 MAPK signaling pathway

To evaluate enteric glia reactivity, we applied ATP, a trigger of purinergic signaling and an inflammatory mediator, to primary msEGC in culture. Our msEGC cultures were highly enriched in

GFAP-expressing cells (mean, $86 \pm 2\%$, Fig EV1A) that also showed Sox10 and S100β immunoreactivity (Fig EV1B) representing the main EGC phenotype seen *in vivo* (Boesmans *et al*, 2015) and enriched glial marker expression (Appendix Fig S1A).

RNA-Seq analysis of the glial transcriptome identified the unique gene dysregulation profile induced by ATP in msEGCs. We found profound changes in msEGC gene expression with 2,027 up-regulated and 2,218 down-regulated genes after ATP stimulation (fold change $\geq 1.5$; *P*-value: $< 0.05$, Fig 1A and principal component analysis shown in Fig EV1C). Therefore, ATP caused up-regulation in 10% and down-regulation in 11% of total glial transcriptome. Induction of genes, known to be expressed in direct response to ATP, including members of the regulator of calcineurin (*RCAN*) (Canellada *et al*, 2008) and *FOS* (Pacheco-Pantoja *et al*, 2016) gene families were confirmed by both RNA-Seq and qPCR (Figs 1H and EV1D). Gene ontology (GO) enrichment analyses demonstrated a general glial activation in ATP-treated msEGCs showing enriched genes for "ATP binding" and "glial proliferation" (Fig 1B and Appendix Fig S1B). Importantly, challenge with ATP induced genes involved in the regulation of cell motility, cytokine response genes (Fig 1C and D and Dataset EV1) and the mitogen-activated protein kinase (MAPK) pathways (Fig 1E and Dataset EV1) underlining the transition of msEGCs to an activated immune phenotype, also referred to as "*gliosis*". The term gliosis is commonly used to describe reactive astrocytes, the CNS counterparts to EGCs. Transcriptionally, gliosis is characterized by the up-regulation of a particular gene set, including, inflammatory response genes. To analyze the reactivity of the EGCs, we created a new GO term for gliosis based on all recent reports discussing CNS gliosis induced by inflammatory stimuli *in vivo* and *in vitro* (Zamanian *et al*, 2012; Hara *et al*, 2017; Liddelow *et al*, 2017; Fujita *et al*, 2018; Mathys *et al*, 2019; Rakers *et al*, 2019; Schirmer *et al*, 2019). Notably, we found that many gliosis-related genes are also regulated in ATP-activated msEGCs (Fig 1F, Appendix Fig S1C and Dataset EV1). Quantitative PCR confirmed the up-regulation of key markers of gliosis including *GFAP* and *NESTIN* (Fig 1G) as well as inflammatory mediators like *CXCL2* and *IL-6* (Fig 1I and J). The latter has been shown to be an important EGC-derived cytokine released upon IL-1β stimulation during surgical trauma (Stoffels *et al*, 2014). Our data confirmed a robust dose-dependent and statistically significant increase in the levels of IL-6, in both mRNA and protein (Fig 1J and K) upon ATP stimulation, indicating a prominent role of IL-6 in activated EGCs, subsequently making it a reliable marker in enteric gliosis and a central part of our further investigations. To efficiently analyze and describe the glia transformation to a reactive phenotype, we chose six targets; *NESTIN* and *GFAP*, two structural glia genes; *IL-6* and *CXCL2*, two inflammatory mediators and *FOSb* and *RCAN*, two transcriptional targets of ATP signaling, as a reliable gliosis marker panel developed from our in silico-based method to further evaluate purinergic enteric gliosis in subsequent studies.

Given that ATP treatment led to an activation of MAPK pathways (Fig 1E), we investigated the involvement of p38-MAPK, an important molecular switch of inflammatory pathways and astrogliosis in the central nervous system (Roy Choudhury *et al*, 2014). ATP was shown to elevate phospho-p38-MAPK protein (Fig EV1E) which is strongly localized in the nucleus of GFAP-positive msEGCs, absent in untreated msEGCs (Fig EV1F). Furthermore, ATP-induced IL-6 protein release was dose-dependently suppressed using the

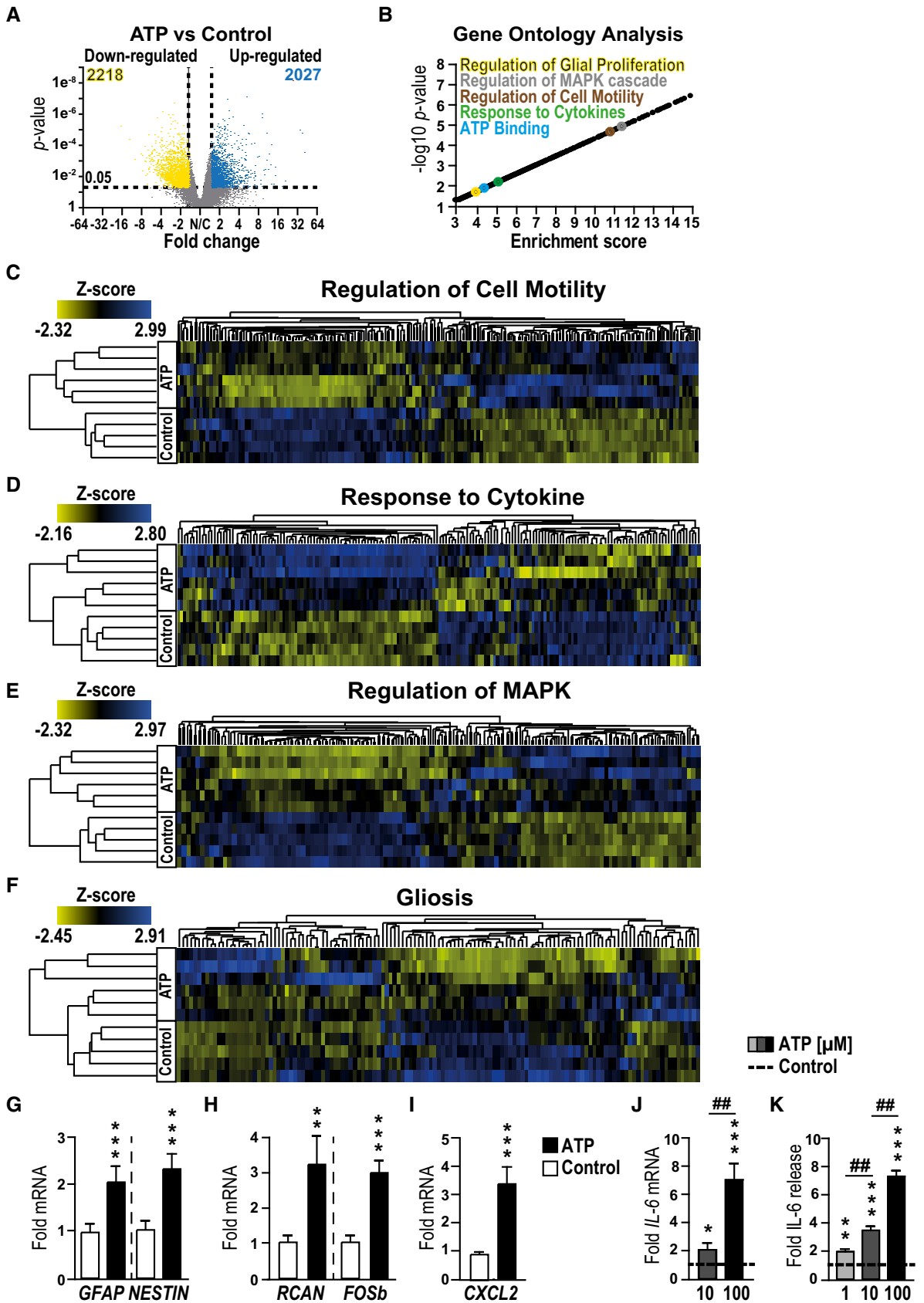

**Figure 1.**

**Figure 1. ATP induces a gliosis in msEGCs.**

A       Volcano plot showing significantly regulated genes between control and ATP-treated msEGCs.
B       Visual representation of GO terms associated with enriched genes in ATP-treated msEGCs compared to control.
C–F    Heat maps of indicated GO terms in ATP-treated msEGCs compared to control.
G–I    qPCR analysis of indicated gliosis genes in ATP-treated EGCs.
J       qPCR analysis of *IL-6* in msEGCs that were treated for 6 h with ATP.
K       IL-6 protein levels in supernatants from msEGCs collected after 24 h treatment with ATP.

Data information: In (A), data are shown as fold change > 1.5, *P*-value < 0.05; ($n$ = 5 for untreated and $n$ = 6 for ATP-treated EGCs). In (G–K), data are shown as fold change + SEM. (G–I) $n$ = 6–9, msEGCs. (J) $n$ = 4–9, msEGCs. (K) $n$ = 6–18, msEGCs. In (A–I): ATP concentration was 100 μM. In (J, K), ATP concentration was 1, 10, or 100 μM. Statistics were performed by applying unpaired Student's *t*-test (G–K) and/or one-way ANOVA with a subsequent Bonferroni test (J and K). In (A) a limma-trend pipeline model and in (B) the Fishers exact test were performed. * indicates significance to control, and [#] indicates significance to ATP treatment with *$P$ < 0.05, **/[##]$P$ < 0.01 and ***$P$ < 0.001.

p38-MAPK-inhibitor SB203580 (Fig EV1G); qPCR confirmed the transcriptional reduction of IL-6 and other gliosis markers like *GFAP*, *CXCL2,* and *RCAN* (Fig EV1H).

Altogether, our data demonstrate that EGC gliosis can be triggered by ATP and induction of enteric gliosis depends on activation of the p38-MAPK signaling pathway.

### P2X receptors mediate the ATP-triggered IL-6 release from msEGC

ATP can be enzymatically dephosphorylated and is, together with its metabolites ADP and adenosine, able to signal via multiple purinergic receptors. Those receptors, broadly divided into the P2X, P2Y and P1 classes (Galligan, 2008), make ATP's signaling repertoire rather complex. Many of these receptor subtypes have been identified in enteric glia, although their role in normal or disease states remains unclear (Ochoa-Cortes *et al*, 2016; Grubišić & Gulbransen, 2017; Gulbransen & Christofi, 2018).

As a starting point to pinpoint the purinergic receptor subtype(s) involved in enteric gliosis, we performed pharmacological screening with various agonists and antagonists of the purinergic signaling system. In our analysis, adenosine failed to stimulate IL-6 release from msEGCs, suggesting that the P1 class is not involved in the ATP-induced phenotype (Fig EV2A). Next, we tested the non-selective P2-class antagonist suramin that showed a blockade of ATP-dependent IL-6 release in a concentration-dependent manner (Fig 2A). Similar results were observed with PPADS, another P2 antagonist (Fig EV2B). Additionally, the degradation resistant ATP isoform and P2 agonist ATPγS, dose-dependently increased the IL-6 release with comparable or even stronger efficacy than ATP itself (Fig 2B) and induced the expression of established gliosis marker genes (data not shown). These findings indicated that ATP, but not ADP, AMP, adenosine or inosine are likely involved in ATP-induced EGC gliosis, thereby limiting the involved receptors to members of the P2 class.

Next, a P2 receptor mRNA expression profile in msEGC was determined in cells isolated from GFAP[cre] x Ai14[fl/wt] mice, expressing tdTomato in all GFAP[+] cells. Cells were either directly sorted upon ME digestion or sorted upon an intermediate cell culture period (Fig EV2D, F and G). Highly increased gene expression of *GFAP* and *Sox10* in tdTomato[+] compared to tdTomato[−] cells confirmed a successful enrichment of msEGC in both procedures (Fig EV2E). Comprehensive gene expression analyses of purinergic receptors in isolated EGC showed a distinct and comparable *ex vivo* and *in vitro* gene expression profile with three P2X receptor genes reaching the highest levels, exceeding not only P1

(Appendix Fig S2A and B), but also P2Y expression levels by several times. Accordingly, we directed our focus toward these P2X receptors expressed in enteric glia in the order P2X7 > P2X4 > P2X2 (Fig EV2H–K).

### P2X2 receptors mediate the ATP-triggered EGC gliosis

P2X7 has been shown to be prominently involved in inflammatory processes. However, neither blockade of P2X7 receptors (Fig 2C) nor its activation with selective agonists (Appendix Fig S2C) could influence IL-6 release in EGCs. While we made the same observation with a P2X4 antagonist, P2X2 antagonism by PSB-1011 (Baqi *et al*, 2011) significantly decreased the ATP-triggered IL-6 release by around 40% (Fig 2C) and demonstrated a dose-dependent inhibitory effect (Fig EV2C). Another P2X2 antagonist (PSB-0711) tested at a lower ATP concentration supported the findings of PSB-1011 (Fig EV2C). PSB-1011 treatment also reduced the gliosis-triggered mRNA expression of *IL-6*, *GFAP*, and *RCAN* (Fig 2D). The absence of cleaved-caspase 3 in msEGCs and no changes in the MTT signal after PSB-1011 treatment confirmed that the reduced IL-6 release and gene expression was not due to apoptosis or reduced cell viability (Appendix Fig S2D and E). To reinforce the pharmacological data with a P2X2 antagonist, we were able to confirm a strong P2X2 immunoreactivity in msEGCs (Fig 2E) with a specific P2X2 signal (Appendix Fig S2F) and used a genetic approach with P2X2-siRNA to block the response. The efficiency of the P2X2 knockdown was confirmed on mRNA (Fig 2F) and protein level (Fig EV2L and M) and it reduced ATP-induced gliosis marker expression on mRNA (Fig 2F) and protein level (Fig 2G).

Together, these pharmacological and siRNA data prove that ATP activates a P2X2 receptor to trigger msEGC gliosis. P1, P2Y or any other highly expressed P2X receptors are not likely to be involved in ATP-triggered EGC gliosis.

### Surgical bowel manipulation in a mouse model of postoperative ileus induces ATP-target gene expression and enteric gliosis

The next series of experiments were performed to investigate the role of ATP on EGC in an *in vivo* model of surgical intestinal manipulation (IM, Appendix Fig S3A) that induces enteric neuroinflammation in the ME and subsequently results in impaired gastrointestinal motility, clinically known as POI. Previous work of our group demonstrated that EGC are involved in POI pathogenesis (Stoffels *et al*, 2014). Hallmarks of POI are an increased IL-6 release (Wehner *et al*, 2005), the infiltration of blood-derived immune cells into the

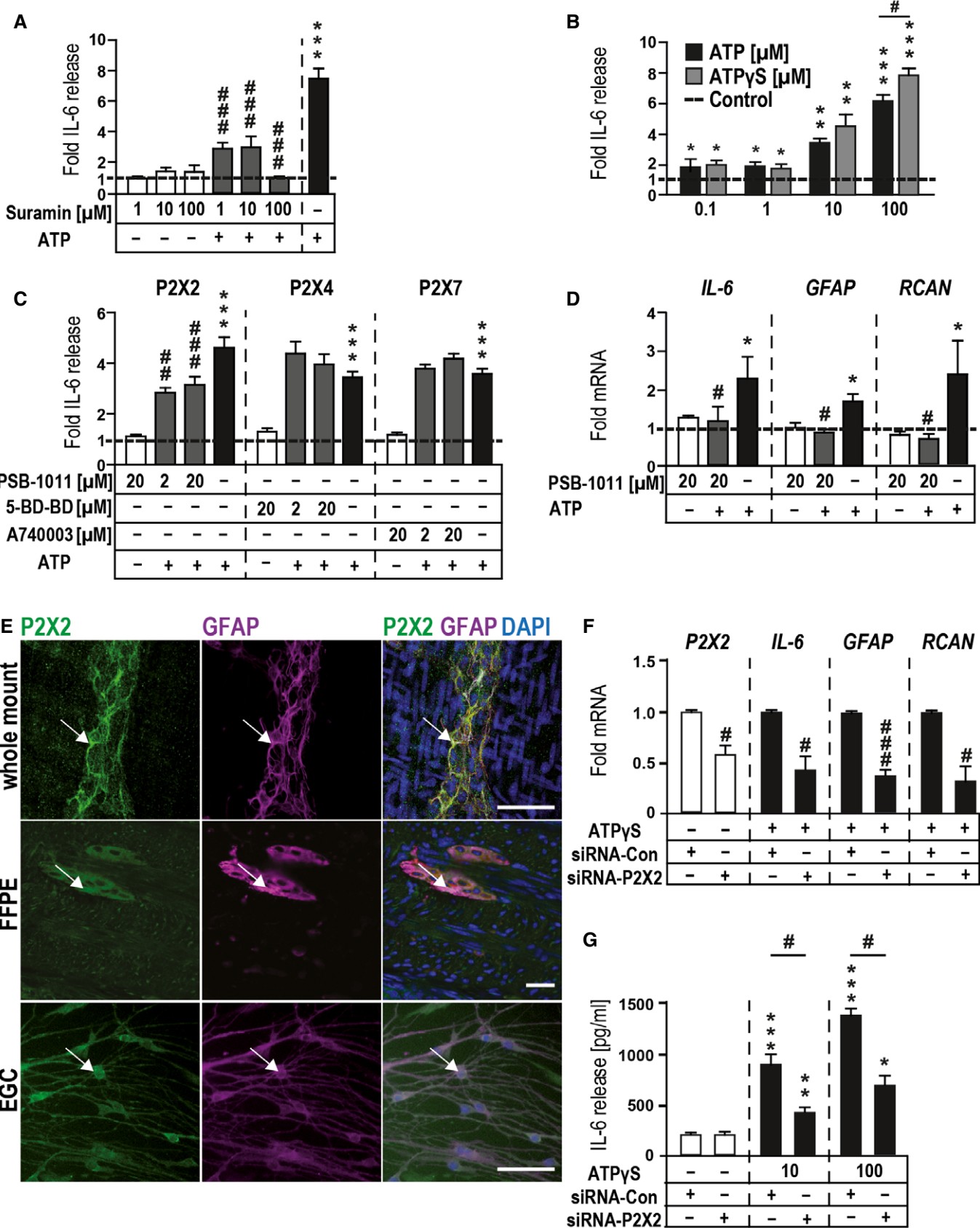

Figure 2.

**Figure 2. ATP-induced gliosis is mediated by p38-MAPK and selective purinergic signaling.**

A Effect of P2 receptor antagonism on ATP-induced IL-6 release. Cells were treated with P2 antagonist suramin (1, 10, and 100 μM) alone or together with ATP (100 μM) for 24 h.

B ATP-induced IL-6 release in msEGCs measured by ELISA. Cells were treated with the indicated concentrations of ATP and ATPγS for 24 h.

C Effects of P2X antagonists on ATP-induced IL-6 release. Cells were treated for 24 h alone or together with ATP (100 μM) in absence or presence of P2X2, P2X4, and P2X7 antagonists PSB-1011, 5-BD-BD, and A740003, respectively.

D P2X2 antagonism of ATP induced mRNA expression of *IL-6*, *GFAP*, and *RCAN* by qPCR in msEGCs. Cells were treated with the P2X2 antagonist PSB-1011 (20 μM) alone or together with ATP (10 μM) for 6 h.

E Representative confocal images of P2X2 (green)- and GFAP (violet)-positive msEGCs *in vivo* and *in vitro*. White arrows mark double-positive (white) cells. Scale bar 50 μm.

F P2X2-siRNA reduces P2X2-mRNA and dampens the gliosis gene expression after ATPγS (100 μM) treatment for 6 h.

G P2X2-siRNA reduces IL-6 release after ATPγS treatment (10, 100 μM) for 6 h.

Data information: In (A–D and F), data are shown as fold induction + SEM and (G) as IL-6 pg/ml + SEM, (A): $n = 10$–15, msEGCs; (B): $n = 3$–15, msEGCs; (C): $n = 8$–17, msEGCs; (D): $n = 4$, msEGCs; (E): $n = 3$–5, msEGCs; (F): $n = 3$–5, msEGCs; (G): $n = 3$–5, msEGCs. Statistics in (A–D, F, G) were performed by applying unpaired Student's *t*-test and/or one-way ANOVA with a subsequent Bonferroni test. * indicates significance to control, and [#] indicates significance to ATP treatment with */[#]$P < 0.05$, **/[##]$P < 0.01$, and ***/[###]$P < 0.001$.

manipulated ME, activation of resident macrophages, and subsequent impairment of gastrointestinal (GI)-transit (Appendix Fig S3B–D). To evaluate whether ATP is involved in the pathogenic mechanism of POI we first sought to measure ATP release in the peritoneal cavity in the lavage fluid to test whether it was elevated after IM. Here, we detected a time-dependent increase of ATP release after IM (Fig 3A). Simultaneously, gene expression of the ATP degrading enzymes *ENTPD2*, *ENTPD8* and *CD73* decreased, indicating a shift in postoperative ATP metabolism that may favor higher extracellular ATP levels that can further activate EGCs (Fig EV3A). In the early phases of POI, immunohistochemistry showed FOSb[+]/Sox10[+] cells in enteric ganglia (Fig EV3B) and strong induction of *FOSb* and *RCAN* gene expression by qPCR (Fig 3H) indicating active ATP signaling in EGC.

RNA-Seq analysis on murine ME specimens isolated 3 or 24 h after IM or from naïve animals showed substantial differences in the gene expression patterns between all tested groups (Figs 3B and EV3C and Dataset EV2). GO enrichment indicated significant regulation of gliosis-associated genes 3 and 24 h after IM and showed similar activation patterns between the postoperative ME and ATP-treated msEGC cultures with alterations in "MAPK", "cell motility", "inflammatory response signaling" and genes involved in "glial development and proliferation" (Fig 3C). Using the previous gliosis gene panel, we confirmed the induction of enteric gliosis during POI progression as demonstrated by gene up-regulation (Fig 3D and Appendix Fig S3F). The strongest response occurred at IM24h. Moreover, a Venn diagram of the gliosis genes displays the similarity of regulated genes *in vivo* and *in vitro* with shared up and down-regulated genes (Appendix Fig S3E), indicating a similar glial activation pattern (Fig 3E). This POI phenotype was confirmed by qPCR showing elevated levels of our established gliosis marker: *GFAP*, *NESTIN*, *IL-6*, *CXCL2*, *FOSb*, and *RCAN* at 3 and 24 h after IM (Fig 3F–H) with a similar increase in IL-6 and CXCL2 protein levels (Fig EV3D).

Next, we analyzed glial proliferation and morphology by immunohistochemistry, defining two hallmarks of gliosis (Buffo *et al*, 2008; Pekny & Pekna, 2016). During disease progression within 24 h the Ki67[+]/Sox10[+] EGC numbers increased up to 10-fold and the glial morphology gains a more complex "branchwood" in myenteric ganglia, revealing a postoperative change in the EGC phenotype (Figs 3I and EV3E). Furthermore, using once more the tdTomato[+]-glia-reporter mouse in our POI model, we were able to

gain further insights into the glial expression profile after IM (Fig 3J). Interestingly, the expression analysis of gliosis markers in EGC showed a different pattern, as previously seen in whole ME tissue after IM. In line with the proliferation, the highest gene expression was detected at IM24h with impressive levels of up-regulation in gliosis genes, including *GFAP*, *NESTIN*, *IL-6*, *CXCL2*, *RCAN*, and *FOSb* mRNA (Fig 3K–M). These results provided us with an *in vivo* insight of enteric gliosis during acute inflammation demonstrating again the immune response of EGCs in POI, a post-surgical intestinal inflammatory disease.

### P2X2 antagonism by ambroxol prevents enteric gliosis

Our next series of experiments tested whether P2X2 antagonism can attenuate or prevent enteric gliosis and improve motility in our mouse model of POI. While the P2X2 antagonists PSB-1011 and PSB-0711 were not applicable for in *in vivo* usage the prospect of future clinical application of a P2X2 antagonist drug to regulate enteric gliosis and protect against developing POI, prompted us to perform a drug library screening to identify a clinically feasible P2X2 antagonist. In this drug screening, ambroxol (Appendix Fig S4A) was identified, as it showed a significant inhibition of ATP-induced calcium influx in 1321N1 astrocytoma cells transfected with the human P2X2 receptor (IC$_{50}$: $5.69 \pm 1.06$ μM), but not in cells transfected with P2X1, P2X3, P2X4, or P2X7 receptors (Fig EV4A). Thus, ambroxol was characterized as a potent P2X2 receptor antagonist with selectivity for P2X2 vs the other P2X receptor subtypes.

After confirming that ambroxol inhibited ATP-triggered gliosis marker up-regulation in msEGCs similar to the before used P2X2 antagonists without any apoptotic effects (Appendix Fig S4B–D), we tested ambroxol as a prophylactic treatment in the POI animal model (Fig 4A). Interestingly, we found a reduced postoperative ME gene expression of ATP-gliosis targets *FOSb* and *RCAN* (Fig 4B) as well as *GFAP* and *NESTIN* (Fig 4C) 3 and 24 h after surgery. In line with these findings, postoperative levels of IL-6 and CXCL2 (Figs 4D and EV4B) were also reduced. Simultaneously, other prototypical pro-inflammatory markers in POI, such as *CCL2* or *TNF-α*, were not affected by ambroxol treatment (Fig EV4C). Antagonism with ambroxol had a discrete influence on pro-inflammatory signaling pathways. To validate direct effects of the P2X2 antagonism on EGCs, we quantified glia proliferation at IM24h for both treatment groups. In comparison, ambroxol dampened the proliferation rate

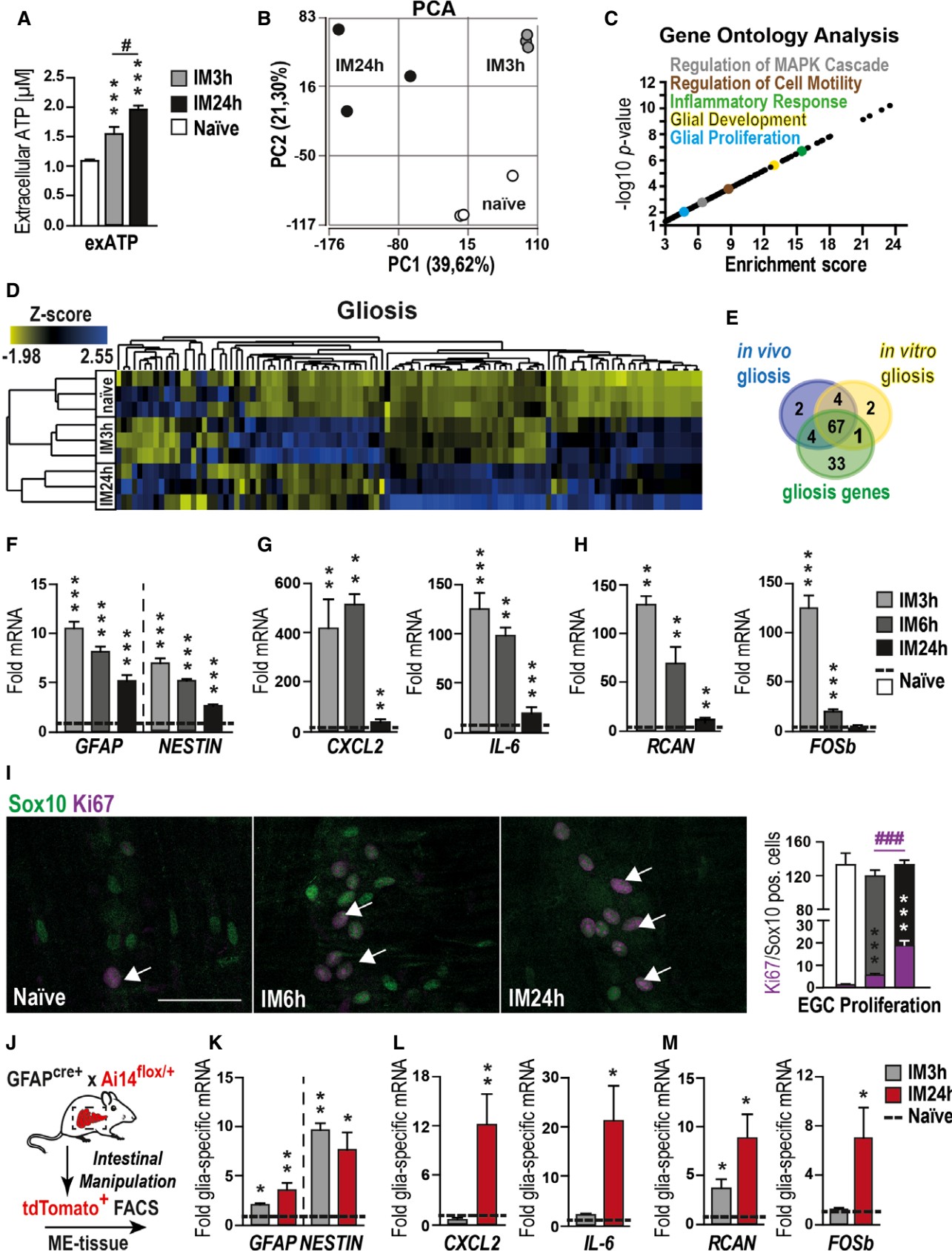

**Figure 3.**

**Figure 3. An ATP-induced enteric gliosis in postoperative bowel inflammation.**

A   ATP measurement at the indicated postoperative time points in peritoneal lavages of mice that underwent intestinal manipulation (IM) or were naïve.
B   PCA plot of gene expression from POI mice at different disease stages and naïve mice; $n = 3$ for each group.
C   Visual representation of P-value ($-\log_{10}$) against fold enrichment of GO terms associated with enriched genes in mice that underwent IM.
D   Heat map of enriched genes connected with GO term gliosis in mice that underwent IM or in naïve animals.
E   Venn diagram of gliosis genes expressed in vitro and in vivo.
F–H qPCR analysis of indicated gliosis marker in mice that underwent IM ($n = 7$).
I   Histological analysis of EGC proliferation in vivo. Representative confocal images and quantification show Sox10 (green)- and Ki67 (violet)-positive EGCs (white arrows) in the small bowel muscularis externa at naïve, IM6h and 24 h. Scale bar 50 μm.
J   Experimental setup for glia-specific RNA analysis. GFAP-cre$^{+}$ × Ai14-floxed mice underwent IM, and small intestine ME was digested and sorted for tdTomato by FACS to provide glia-specific RNA for qPCR measurements seen in (K-M).
K–M qPCR analysis of indicated gliosis markers in td + glia cells from naïve mice and mice that underwent IM ($n = 5$–7, POI mice).

Data information: In (A), data are represented as mean + SEM; $n = 7$–12, POI mice. In (F–H and K–M), data are represented as fold induction + SEM. In (I), data are represented as mean of double-positive cells per total Sox10-positive cells + SEM; 9–20 whole mount specimens per conditions; $n = 8$, POI mice per group. Statistics were performed by applying unpaired Student's t-test (A, C, F–I, K–M). In (C), the Fisher's exact test was performed. * indicates significance to naïve animals, and $^{\#}$ indicates significance to the indicated time point with $*/^{\#}P < 0.05$, $**P < 0.01$, and $***/^{\#\#\#}P < 0.001$.

by almost 50%, indicating reduced glial activation during POI (Fig 4E). Finally, we examined a modulatory effect on immune cell infiltration by ambroxol and discovered that ambroxol significantly reduced the number of monocytes (CD45$^{+}$/Ly6C$^{+}$/CX3CR1$^{-}$) and resident (CD45$^{+}$/Ly6C$^{-}$/CX3CR1$^{+}$) or monocytes derived (CD45$^{+}$/Ly6C$^{+}$/CX3CR1$^{+}$) macrophages (CD45$^{+}$/Ly6C$^{+/-}$/CX3CR1$^{+}$) in the manipulated ME at 3 h (~ 70% reduction, MPO-histology) and 24 h (~ 50% reduction, MPO-histology and FACS; Figs 4F and G, and EV4D). In addition, histological analysis of the localization of CX3CR1$^{+}$-cells in context to EGCs showed fewer macrophages surrounding ganglia in the ambroxol-treated group (Fig EV4E). Functionally, ambroxol led to a significant improvement in postoperative gastrointestinal transit time (geometric center ambroxol: $6.8 \pm 0.4$ vs vehicle: $4.1 \pm 0.4$, Fig 4H). These data indicated that P2X2 antagonism is an effective strategy to attenuate gliosis to improve clinical symptoms in the mouse POI model and makes ambroxol a relevant P2X2 antagonist drug candidate for future therapeutic approaches.

**Human enteric gliosis is blocked by P2X2 antagonism**

To determine whether findings in the mouse are translatable to human, we carried out additional experiments in human specimens from surgical patients who underwent pancreatectomy, a procedure in which intense IM is an unavoidable consequence, enabling the collection of small bowel specimen at two different intraoperative time points (Fig 5A). In this series of experiments, we analyzed the expression of gliosis markers and ATP-dependent genes in ME samples of jejunum specimens. Similar to our previous analyses, we performed RNA-Seq analysis in a limited set of patient ME samples to gain more insight into the glial activation status. The PCA showed a distinct difference between the early and late samples (Fig EV5A) and the consecutive GO analysis pointed to a similar activation pattern as previously shown in the murine system with enriched genes for "MAPK cascade", "regulation of cell motility", "immune response" and "glial proliferation" (Fig 5B, Dataset EV3). Consistently, the gliosis panel showed differential gene expression between late and early collected specimens, providing evidence for inflammation-induced enteric gliosis during surgery (Fig 5C). A Venn diagram comparing the murine and the human enteric gliosis genes visualized 37 shared genes between these species (Fig 5D). To strengthen the RNA-Seq data, we validated our gliosis marker

panel in 13 more patients. Compared to the foremost collected ME specimens, specimens collected at the later time point showed a strong increase in IL-6-protein (Fig EV5B) and mRNA levels (Fig EV5C) in concordance with the induction of other gliosis markers (Fig EV5C).

To determine the involvement of glia in inflammatory processes occurring after IM in patients, we subjected human EGCs, isolated and purified from single human myenteric ganglia dissociated from ME specimen (hEGC) to ATP and ATPγS in vitro stimulation according to our established protocols (Ochoa-Cortes et al, 2016). Both treatments induced a more than 12-fold increase in IL-6 protein release after 6 h, which dropped to a still significant two-fold induction after 24 h (Fig 5E). Interestingly, naïve hEGC exhibited a dose-dependent increase in IL-6 release upon stimulation with ARL67156, a selective ectonucleoside triphosphate diphosphohydrolase (ENTPDase) inhibitor (Fig 5F) indicating that ATP levels are tightly regulated by ENTPDases and that their inhibition creates a high-enough endogenous ATP concentration to activate hEGC in vitro.

As purinergic activation in hEGC showed similarities to the mouse data, we also investigated human P2X2 signaling. Immunofluorescence microscopy revealed strong P2X2-immunoreactivity in s100β$^{+}$ hEGCs in myenteric ganglia in intact surgical tissues (Figs 5G and EV5D) and more than 80% of cultured hEGCs (Fig EV5E). Additionally, P2X2 antagonism by PSB-1011 blocked the ATP-triggered IL-6 release in hEGCs (Fig 5H). The antagonistic specificity for the human system of our tested P2X2 inhibitors was also confirmed in HEK293-P2X2 sniffer cells expressing a mutant human P2X2 receptor resistant to desensitization. This process leads to continuous elevation of free intracellular calcium levels upon ATP stimulus in a dose-dependent manner (Appendix Fig S5A). ATP-induced calcium levels were independently reduced by the P2X2 antagonists PSB-1011 (Fig 5I) and PSB-0711 (Fig 5J and Appendix Fig S5B) verifying that these antagonists are able to block responses mediated via hP2X2 in vitro. Based on the promising results of ambroxol as a therapeutic in mice, we utilized it further in an experimental approach in hEGC. Ambroxol treatment inhibited dose-dependently ATP-induced calcium influx in HEK293-P2X2 sniffer cells (Appendix Fig S5C) and blocked ATP-triggered IL-6 release in hEGC (Fig 5K).

Together, these data confirm the relevance of ATP-triggered P2X2 signaling in hEGC gliosis and identify ambroxol as a novel

P2X2 antagonist in the human system. To finally confirm the role of ambroxol in human specimens, freshly isolated human full-thickness jejunal samples, underwent an *ex vivo* mechanical alteration in

the presence or absence of ambroxol (Fig EV5F). All relevant gliosis genes followed up in this study, were significantly down-regulated in the ambroxol-treated extracorporally manipulated jejunal ME

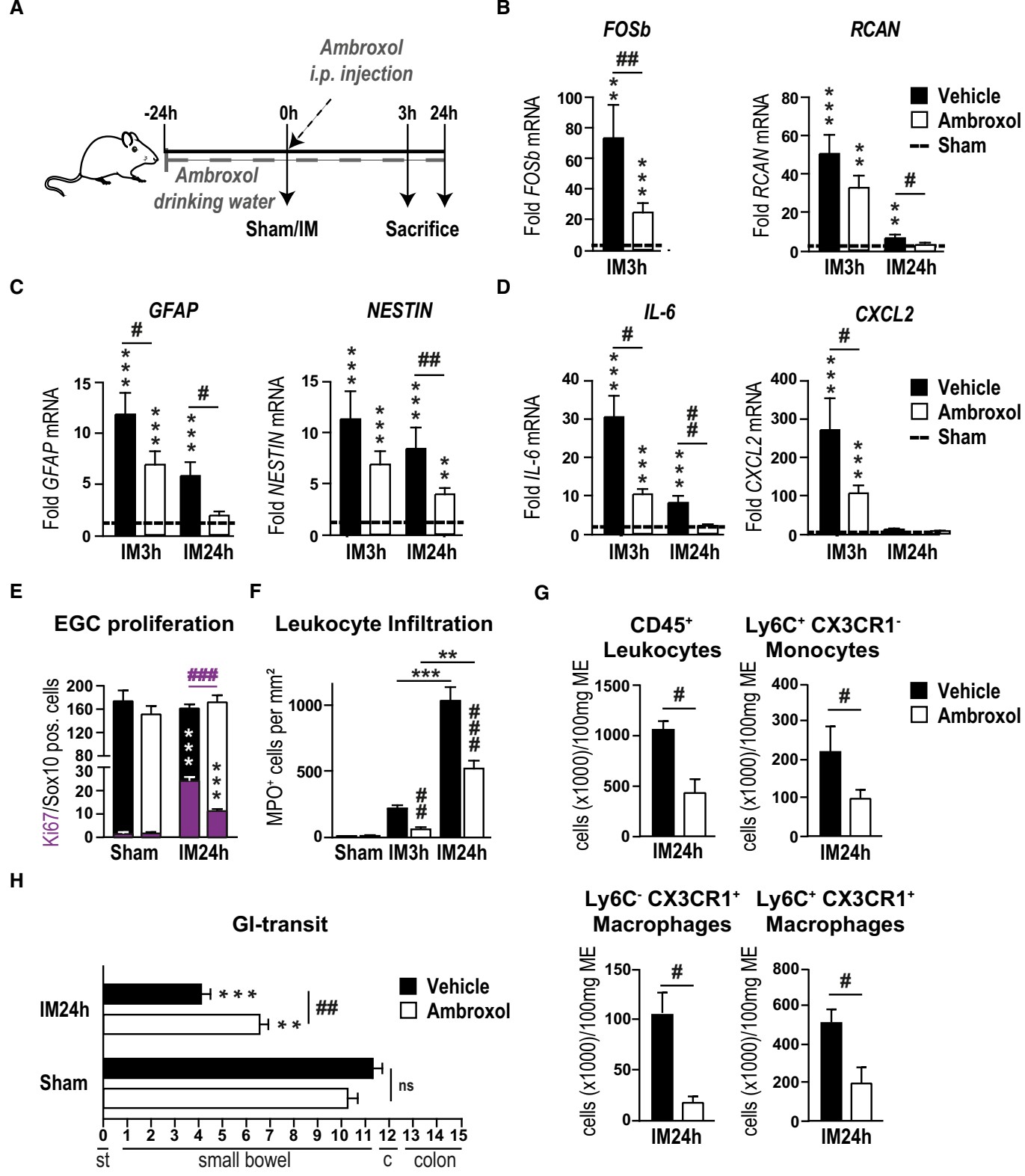

**Figure 4.**

◀

**Figure 4. The novel P2X2 antagonist ambroxol prevents IL-6 release from EGCs.**

A Ambroxol treatment scheme. Mice were treated with ambroxol or vehicle and underwent a sham operation (laparotomy) or intestinal manipulation (IM). Small bowel muscularis externa (ME) was isolated and analyzed 3 or 24 h after surgery.

B–D Postoperative gene expression analyses of indicated gliosis marker in the ME.

E Histological analysis of EGC proliferation by quantification of Ki67 (violet)- and Sox10 -positive EGCs at IM24h. Mice were treated as seen in (A).

F Histological counting of infiltrating myeloperoxidase (MPO)-positive leukocytes in the postoperative ME ($n = 6$–8).

G FACS analysis of infiltrating cells in the ME of mice treated with ambroxol or vehicle. CD45, Ly6C, and CX3CR1 were used to distinguish resident macrophages (CD45$^+$/Ly6C$^-$/CX3CR1$^+$), infiltrating monocytes (CD45$^+$/Ly6C$^+$/CX3CR1$^-$) and infiltrated monocyte-derived macrophages (CD45$^+$/Ly6C$^+$/CX3CR1$^+$).

H Postoperative *in vivo* GI transit measurement in mice treated with ambroxol or vehicle.

Data information: In (B–D), data are represented as fold change + SEM vs the sham groups ($n = 6$–8, POI mice per group). In (E), data are represented as mean of double-positive cells per total Sox10-positive cells + SEM; six whole mount specimens per conditions; $n = 11$ mice per IM group and $n = 3$ per sham group. In (F), data are presented as mean + SEM MPO$^+$ cells/mm$^2$ small intestine ME tissue. In (G), data are presented as cells per 100 mg ME tissue + SEM $n = 3$–5 POI mice per group. In (H), data are presented as mean + SEM; $n = 12$ mice per group. Statistics were performed by applying unpaired Student's $t$-test and one-way ANOVA with a subsequent Bonferroni test (B-H). * indicates significance compared to sham animals, and $^\#$ indicates significance between vehicle and ambroxol treatment with */$^\#P < 0.05$, **/$^{\#\#}P < 0.01$, and ***/$^{\#\#\#}P < 0.001$.

(Fig 5L–N) indicating that ambroxol treatment is sufficient to attenuate trauma-induced gliosis.

While future studies, particularly clinical trials, must prove if ambroxol also prevents surgery-induced gliosis in patients, our *ex vivo* human data corroborate its role in dampening gliosis and ATP-driven inflammatory processes and provide proof of concept for translatability of findings on P2X2 from mice to humans.

# Discussion

In this study, we aimed to define and better understand the reactive glial phenotype of the enteric nervous system induced by ATP and clarifying its role, beneficial or adverse, concerning inflammation-induced motility disorders, and in particular POI. In this regard, it has been previously shown by our group that activated EGCs contribute to the disease progression in POI (Stoffels *et al*, 2014). However, to this day, the significance of glial reactivity in the pathogenic mechanism remained unclear. We provide evidence to support the hypothesis that surgical manipulation and trauma triggers ATP release that drives enteric gliosis and intestinal inflammation leading to impaired motility and POI.

In general, reactive changes of glial cells are a hallmark of "gliosis" that is known to be induced in the CNS by numerous pathological conditions including traumatic (Andersson *et al*, 2011) or ischemic insults (Roy Choudhury *et al*, 2014) and in neurodegenerative diseases (Pekny & Pekna, 2014). As a predefined-GO term for gliosis did not exist and in order to achieve a more integrative description of a reactive EGC phenotype, we created a non-exclusive list of published genes regulated in reactive astrogliosis (Zamanian *et al*, 2012; Hara *et al*, 2017; Liddelow *et al*, 2017; Fujita *et al*, 2018; Mathys *et al*, 2019; Rakers *et al*, 2019; Schirmer *et al*, 2019). Based on this gene panel, we analyzed EGCs *in vitro* and *in vivo* and termed the glial activation "enteric gliosis". We targeted ATP as a potential trigger mechanism of enteric gliosis. EGCs have been shown to respond to ATP *in situ* (Gulbransen & Sharkey, 2009; Boesmans *et al*, 2019) and *in vitro* (Gomes *et al*, 2009; Boesmans *et al*, 2013) and ATP is a potent trigger of innate immune responses. Increased release of ATP has been observed in multiple acute and chronic inflammatory diseases, including autoimmune diseases (Carta *et al*, 2015), sepsis (Csóka *et al*, 2015), sterile insults (Cauwels *et al*, 2014) and colitis (Grubišić *et al*, 2019). Thus, we

hypothesized that increased ATP levels, which occur as a result of tissue damage to the intestine (Galligan, 2008) or extensive gut manipulation during the operation, might trigger an inflammatory response and activation of EGCs. Our comprehensive RNA-Seq analysis confirmed profound transcriptional changes in EGCs upon direct ATP stimulation visualized by a separation of control and ATP-treated EGCs in a PCA plot. The differences in the severity of the activation by purinergic stimuli most likely come from batch differences of primary cells. Moreover, we detected an EGC profile that is comparable to a reactive astrocyte, the CNS counterpart of EGCs (Grubišić & Gulbransen, 2017). In agreement with the up-regulation of genes involved in MAPK signaling, the main switch in astrogliosis (Roy Choudhury *et al*, 2014), p38-MAPK is also induced in ATP-stimulated EGCs and its blockade completely abrogates enteric gliosis. Interestingly, another of our studies revealed the importance of p38-MAPK (Wehner *et al*, 2009) in intestinal inflammation, although its role as a signaling pathway in enteric gliosis had not been investigated before. From these data, we conclude that ATP stimulation induces phenotypical changes in EGCs that are most precisely described by the term enteric gliosis.

So far, ATP's role in gliosis has not been investigated in detail, but previous studies had speculated on the possible involvement of purinergic receptors (Burda & Sofroniew, 2014). Herein, we identified P2X2, one of the highest expressed P2X receptors in murine and in human EGCs (Liñán-Rico *et al*, 2016), as the purinergic receptor responsible for triggering enteric gliosis upon ATP stimulation. In humans and mice, we confirmed by immunohistochemistry, that hEGCs and glia in the intact human myenteric plexus strongly express the P2X2 receptor. While P1 receptors, in general, could be excluded from the list of involved receptors, other P2 receptors that have not been tested in our study either because of their low expression or due to a lack of selective antagonist/agonist could potentially be involved in ATP-triggered gliosis. However, P2X7, the purinergic receptor with the highest expression in msEGCs and the third highest in hEGC (Liñán-Rico *et al*, 2016) that is also expressed on virtually all immune cell types (Di Virgilio *et al*, 2017), is not involved in the ATP-triggered reactive EGC phenotype.

In contrast, ATP signaling induces neuronal death in models of colitis, another intestinal inflammatory disease, by activating a complex involving P2X7 receptors, Pannexin-1, Asc and caspases (Gulbransen *et al*, 2012). It appears that the glial P2X2 gliosis mechanism is unique to postsurgical inflammation. Interestingly, others

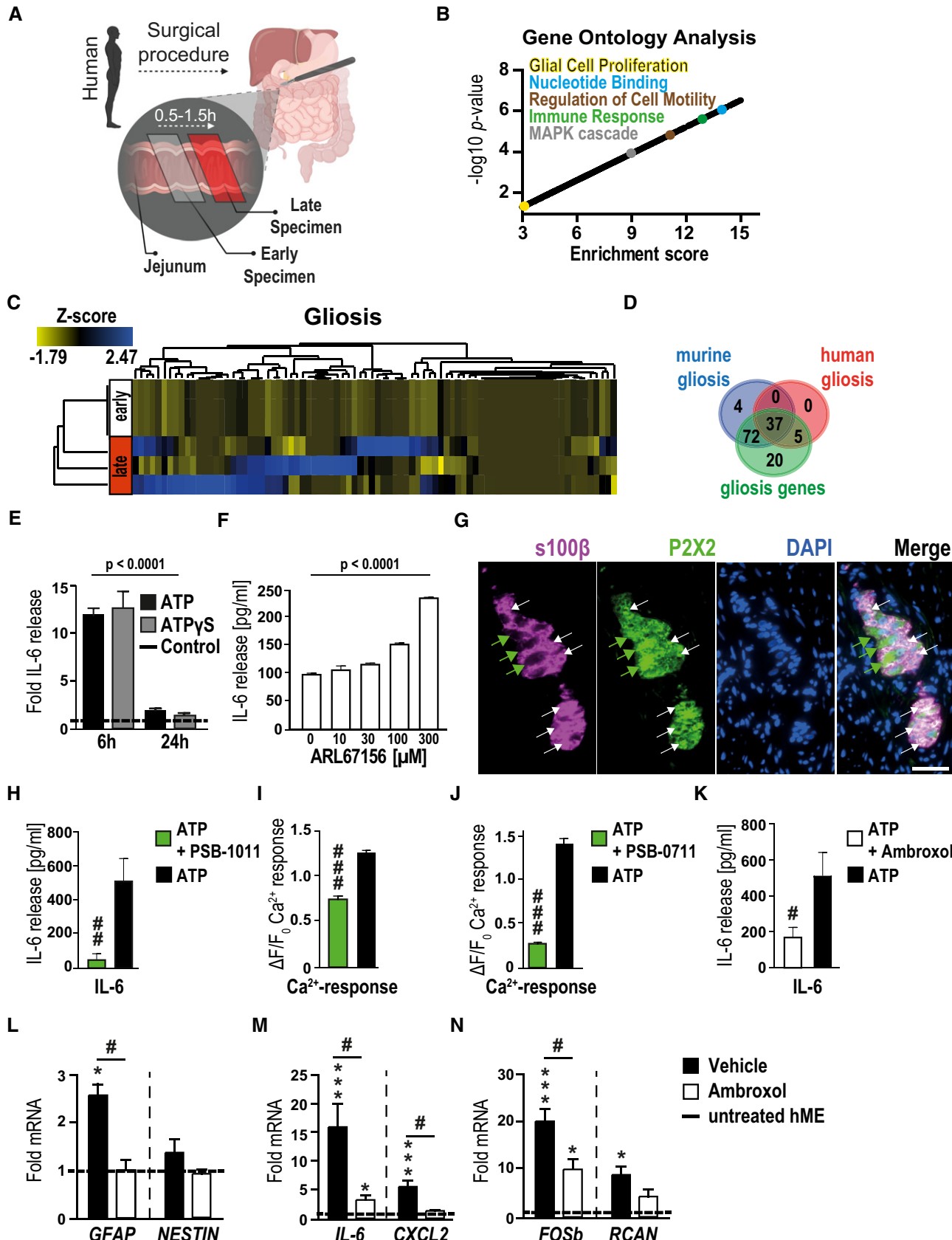

**Figure 5.**

◄

**Figure 5. ATP induces gliosis in hEGC.**

A    Schematic workflow on the collection and processing of surgical specimens collected during a pancreaticoduodenectomy. Samples were provided on ice directly from the operation room, and the ME was separated from lamina propria mucosae.

B    Visual representation of $P$-value ($-\log_{10}$) against fold enrichment of GO terms associated with enriched genes in patient specimen that underwent IM.

C    Heat map of enriched genes connected with GO term gliosis in patient specimens that underwent IM at two time points: early and late; $n = 3$ human patients.

D    Venn diagram of gliosis genes expressed in human (red) and murine (blue) ME tissue.

E    IL-6 protein release in hEGC cultures upon stimulation with ATP (200 µM) or ATPγS (100 µM) after 6-h and 24-h treatment ($n = 4$–$6$, hEGCs).

F    IL-6 protein release in hEGC cultures upon stimulation with NTPDase inhibitor ARL67156 at indicated concentrations; $n = 4$–$6$, hEGCs.

G    Immunofluorescence microscopy revealed P2X2 expression (green) in a majority of s100β$^+$ (violet) hEGCs in intact myenteric ganglia of the human colon. White arrows mark double-positive glia cells, and green arrows mark P2X2-positive neurons. Scale bar, 50 µm.

H    Effect of the P2X2 antagonism on ATP-induced IL-6 release. ELISA measurement of IL-6 in hEGCs upon treatment with P2X2 antagonist PSB-1011 (20 µM) alone or together with ATP (200 µM) treated for 24 h ($n = 6$, hEGCs).

I    PSB-1011 (20 µM) treatment inhibited ATP-triggered calcium responses in HEK cells transfected with P2X2. Data are represented as $\Delta F/F_0 + SEM$; $n = 102$ HEK cells.

J    The P2X2 receptor antagonist PSB-0711 (20 µM) nearly abolished the ATP-triggered calcium response in HEK cells transfected with P2X2. $n = 219$ HEK cells.

K    Ambroxol blocks ATP-induced IL-6 release in hEGCs; protein release measurement by ELISA of IL-6 in hEGCs. Cells were treated with ATP (200 µM) alone or together with ambroxol (20 µM) for 24 h; $n = 6$, hEGCs.

L–N    qPCR analysis of several gliosis and ATP-target genes in the mechanically manipulated surgical specimens incubated in the presence or absence of ambroxol (20 µM) ($n = 7$, 4 human patients).

Data information: In (E, F, H, and K), data are represented as mean IL-6 release + SEM. In (I and J), data are represented as $\Delta F/F_0 + SEM$. In (L–N), data are shown as fold induction + SEM. Statistics were performed by applying unpaired Student's $t$-test and one-way ANOVA with a subsequent Bonferroni test (E and F, H–N), and in (B), the Fisher's exact test was performed. * indicates significance compared to controls, and $^\#$ indicates significance to ATP treatment or between vehicle and ambroxol with $*/^\# P < 0.05$, $^{\#\#} P < 0.01$, and $***/^{\#\#\#} P < 0.001$.

have shown that chronic morphine induced constipation associated with intestinal inflammation involves a glial ATP-connexin signaling pathway (Bhave *et al*, 2017).

In order to analyze the role of enteric gliosis *in vivo* and to access its value as a therapeutic target (Ochoa-Cortes *et al*, 2016; Gulbransen & Christofi, 2018), we chose our established *in vivo* model of intestinal surgical manipulation (IM) that leads to development of POI (Wehner *et al*, 2009; Stoffels *et al*, 2014). Our comprehensive gliosis marker analysis, immune-activation pathway analysis, morphological, proliferation and cytokine release analysis in response to ATP activation allowed us to conclude that abdominal surgery induces enteric gliosis. In line, detection of increased ATP levels in the peritoneal cavity, which is either actively or passively released following cellular damage (Carta *et al*, 2015), clearly indicate that ATP release is part of the postoperative inflammatory cascade (Cauwels *et al*, 2014) and can principally act as a DAMP and an inducer of gliosis. The simultaneous decrease in several nucleotide-catabolizing ecto-enzymes, including CD73 and ENTPD8, shifts the cellular metabolic machinery in favor of elevated extracellular levels of ATP in surgically manipulated bowel to reach levels that are sufficient to induce enteric gliosis *in vivo* via P2X2.

Although the cellular source of ATP remains unknown, the immediate increase in the early phase of POI indicates that it is initially released from resident cells than from infiltrating leukocytes that extravasate into the ME in later stages of POI (Stein *et al*, 2018). These resident cells could be enteric neurons (Gomes *et al*, 2009) or resident macrophages (Riteau *et al*, 2012), as both lie in proximity to EGCs and contain large amounts of ATP that can be actively or passively released upon damage or during inflammation (Oliveira *et al*, 2014). Additionally, the reactive EGCs themselves are another likely local source of ATP release, and we have previously shown that LPS induction of primary hEGCs in culture elevates ATP release by several fold (Liñán-Rico *et al*, 2016). Interestingly, we observed that fewer macrophages surround enteric ganglia upon ambroxol treatment in the POI model. However, it remained unclear if this is due to a general reduction in macrophages or due to a specific

reduction in glial activation. Of interest, a recent story revealed direct effects of activated EGCs on macrophage function (Grubišić *et al*, 2020) substantiating our observation that gliosis is connected to monocyte/macrophage infiltration. Notably, we analyzed previous POI samples of CCR2$^{-/-}$ mice published by Stein *et al* (2018) to address the potential role of the infiltrating immune cells in ATP release and glial activation during the later time points in POI. Intriguingly, glial activation was reduced in CCR2$^{-/-}$ POI mice, indicating that the immune infiltrate maintains enteric gliosis after manifestation of POI and it is likely that this mechanism also depends on ATP release (Appendix Fig S4F). However, as CCR2$^{-/-}$ mice did not show improved motility 24 h after surgery but showed elevated—but reduced—gliosis marker expression, we interpret that the initial gliosis induced by resident cell activation alone is sufficient to trigger POI. Interestingly, CCR2$^{-/-}$ mice show disturbances in the resolution of POI at later time points (72 h) (Farro *et al*, 2017), supporting our hypothesis that the infiltrating monocytes contribute to the regenerative gliosis.

In humans, the same set of gliosis marker genes as in mice was time-dependently up-regulated in jejunal ME tissues of patients who underwent pancreaticoduodenectomy, a surgical procedure that unavoidably involves strong intestinal manipulation. After gut manipulation and trauma in patients, we detected a clear increase in gliosis markers in the late stages of the operation indicating that our murine data on enteric gliosis are translatable to humans. Therefore, we conclude that gliosis is a conserved cross-species mechanism regulating the EGC inflammatory response. Moreover, hEGCs exhibited a comparable receptor expression profile and the same functional dependence on P2X2 as msEGC. Further evidence for a conserved enteric gliosis mechanism across different species originates from our recent *in vitro* studies in hEGCs describing an immune phenotype upon LPS and interferon-γ signaling (Liñán-Rico *et al*, 2016) similar to the one seen in msEGC in the present study.

Our findings lead to a new question: What is the purpose of this conserved gliosis; is the induction necessary for regenerative mechanisms or could an initial blockade lead to a less severe disease

**Figure 6. Purinergic mechanism of enteric gliosis in postoperative ileus.**

Gut surgical manipulation and trauma cause inflammation and increase ATP release that activates P2X2 receptors on enteric glia and induces a reactive glial phenotype termed "enteric gliosis". In the context of inflammation, ATP activates a P2X2/p38 MAPK pathogenic signaling pathway associated with an increased expression in gliosis markers GFAP, NESTIN, and the release of cytokines like IL-6 and CXCL2. Enteric gliosis exacerbates neuroinflammation, contributes to immune cell infiltration and that causes postoperative motility disturbances and POI. P2X2 antagonism by ambroxol prevents the ATP-triggered enteric gliosis and protects against POI.

outcome? In the CNS, gliosis is targeted by therapeutic approaches in neurodegeneration (Colangelo *et al*, 2014) to minimize for example neuronal apoptosis (Livne-Bar *et al*, 2016). In the ENS, there is no evident view on gliosis yet. The general glial activation/proliferation could serve as damage control and/or regeneration boost by assisting the ENS to regenerate faster after an inflammatory strike, but on the other side targeting EGCs in gastrointestinal immune-driven diseases and motility disorders could also be a new and promising therapeutic approach (Gulbransen & Christofi, 2018). Incidentally, the strong up-regulation of glial proliferation in manipulated mice indicated that EGC do participate in ENS regeneration in later stages of POI as already described in colitis-induced chronic intestinal inflammation (Belkind-Gerson *et al*, 2017), in addition to their immune-modulatory role by, e.g., cytokine release. However, to clarify the potential regenerative role of ATP-induced gliosis in POI, future studies need to focus on the regeneration stages of this disorder, i.e., 72 h and later after surgery. (Stein *et al*, 2018).

An internal compound library screening revealed that the antagonist potency of the oral drug ambroxol is comparable to the experimental organic P2X2 antagonist (Baqi *et al*, 2011) and is suitable for an *in vivo* investigation. Interestingly, ambroxol's immune-modulatory effect is already documented in the clinical treatment of airway inflammation, but its mode of action is still unclear (Beeh *et al*, 2008). Indeed, perioperative ambroxol treatment reduced the postoperative gliosis marker increase and EGC proliferation. It also prevented infiltration of monocytes/macrophages and motility impairment in our POI model. The down-regulation of ATP-target genes *FOSb* and *RCAN* implied a direct antagonistic effect on purinergic activation by ambroxol. Importantly, ambroxol prevented induction of *IL-6* and *CXCL2* but no other inflammatory genes, including *TNFα* and *CCL2*. This demonstrates that ambroxol does not function as a general anti-inflammatory drug as previously speculated (Beeh *et al*, 2008) but instead selectively modulates ATP-triggered pathways in EGCs. In support of ambroxol's immune-modulatory action, a recent study using a P2X2/X3 antagonist (gefapixant), showed a down-regulation of airway inflammation, the main target of ambroxol (Zhang *et al*, 2020), upon ambroxol treatment in a cough hypersensitivity syndrome model. Nevertheless,

ambroxol is also known to affect potassium and calcium channels of neurons (Weiser, 2008) that modulate neuronal activity (Magalhaes et al, 2018). As motility is tightly regulated by these channels (Rao, 2020), we cannot exclude any enteric neuronal modulation by ambroxol in parallel to its impact on glial P2X2-dependent signaling.

Given that the purinergic system is a prominent player in inflammation (Burnstock, 2020), selective P2X2 antagonism might be of particular therapeutic relevance in POI or perhaps other gut inflammatory or immune-driven motility disorders. This is supported by animal studies which showed improved clinical symptoms in ambroxol-treated mice in models of neurodegeneration (Migdalska-Richards et al, 2016), neuropathic pain (Hama et al, 2010) and LPS-induced acute lung injuries (Su et al, 2004). Notably, all these diseases lead to activated purinergic signaling (Burnstock, 2020). Finally, the effects of reduced enteric gliosis in humans were supported by ex vivo manipulation of jejunal samples from surgical patients demonstrating a direct inhibition of a mechanically induced gliosis-related gene induction by ambroxol. In line with these findings, previous clinical trials done over 30 years ago with ambroxol also revealed an alleviated motility in the ambroxol-treated group (Germouty & Jirou-Najou, 1987) and ongoing clinical studies for Parkinson's Disease (NCT02941822; (Silveira et al, 2019)) highlight the potential of ambroxol in treating a neurodegenerative disease with a connection to gut inflammation (Villumsen et al, 2019).

Although we used a broad panel of different techniques, we would like to mention that a final validation of our findings would require the use of a glial-specific P2X2 knock out mouse. Furthermore, some of our analyses are based on whole ME tissue samples including multiple non-glia cell types. Even though we validated most of these findings in enriched EGC cultures, a contribution of other cell types in vivo cannot completely be excluded. Nevertheless, future studies should aim to test the clinical efficacy of P2X2 antagonism for the treatment of POI in humans and immune-driven inflammatory diseases, including motility disorders. Ambroxol or yet to be developed highly selective P2X2 antagonists with the implementation of the power of medicinal chemistry and congener drug development (Burnstock et al, 2017) are suggested to represent novel candidate drugs in the pipeline.

Subsumed, we provide evidence that ATP is able to induce a reactive EGC phenotype, increased inflammation and enteric gliosis in vivo in a P2X2 dependent manner. This mechanism proved to represent a pathogenic mechanism of POI, since ambroxol, a novel P2X2 antagonist, was shown to have efficacy in protecting against postoperative bowel inflammation and motility disturbances in mice and humans. Our "purinergic hypothesis of enteric gliosis in POI" is illustrated in Fig 6.

# Materials and Methods

### Murine EGC cultures

Primary enteric glia cell cultures were obtained by sacrificing C57BL/6 mice 8–16 weeks of age, extracting the small intestine and cleansing it with 20 ml of oxygenated Krebs–Henseleit buffer (126 mM NaCl; 2.5 mM KCl: 25 mM NaHCO$_3$; 1.2 mM NaH$_2$PO$_4$; 1.2 mM MgCl$_2$; 2.5 mM CaCl$_2$, 100 IU/ml Pen, 100 IU/ml Strep and

2.5 μg/ml Amphotericin). The small bowel was cut in 3–5 cm long segments and kept in oxygenated ice-cold Krebs–Henseleit buffer. Each segment was then drawn onto a sterile glass pipette and the ME was stripped with forceps to collect muscle tissue for further digestion steps. After centrifugation (300 $g$ for 5 min), the tissue was incubated for 15 min in 5 ml DMEM containing Protease Type1 (0.25 mg/ml, Sigma-Aldrich) and Collagenase A (1 mg/ml, Sigma-Aldrich) in a water bath at 37°C, 150 rpm. The enzymatic digestion was stopped by adding 5 ml DMEM containing 10% FBS (Sigma-Aldrich), centrifugation for 5 min at 300 $g$ and resuspended in proliferation medium (neurobasal medium with 100 IU/Pen, 100 μg/ml Strep, 2.5 μg/ml Amphotericin [all Thermo Scientific], FGF and EGF [both 20 ng/ml, Immunotools]). Cells in proliferation media were kept at 37°C, 5% CO$_2$ for 4 days to promote formation of enteric neurospheres. For experiments, enteric neurospheres were dissociated with trypsin (0.25%, Thermo Scientific) for 5 min at 37°C and distributed at 50% confluency on Poly-Ornithin (Sigma-Aldrich) coated six well plates in differentiation medium (neurobasal medium with 100 IU/Pen, 100 μg/ml Strep, 2.5 μg/ml Amphotericin, B27, N2 [all Thermo Scientific] and EGF [2 ng/ml, Immunotools]). After 7 days in differentiation medium, mature enteric glia cells were treated with ATP (0.1, 1, 10, 100 μM, Sigma), ATPγS (0.1, 1, 10, 100 μM, Sigma), Adenosine (1, 100 μM, Sigma), PPADS (5, 30 μM, TOCRIS), Suramin (1, 10, 100 μM, TOCRIS), A740003 (2, 20 μM, TOCRIS), ambroxol (0.2, 2, 20 μM, TOCRIS), PSB-0711 (2, 20 μM, TOCRIS), PSB-1011 (0.2 2, 20 μM, TOCRIS), 5-BDBD (2, 20 μM, TOCRIS) SB203580 (1, 5, 10 μM, TOCRIS) and further processed for RNA isolation or their conditioned medium used for ELISA or qPCR analysis.

For the siRNA approach, primary msEGCs were differentiated as mentioned above and transfected with a control-siRNA (SIGMA) or P2X2-siRNA (#4390771, Thermo Scientific) lipofectamine (Thermo Scientific) complex for 72 h according to the manufacturer's instructions. Afterward, the transfected cells were treated with ATPγS (10, 100 μM, Sigma) and analyzed by qPCR and ELISA. For Western Blotting, primary msEGCs were lysed in RIPA buffer, centrifuged at maximum speed for 20 min and prepared with loading buffer (Bio-Rad). All samples were processed with the Bio-Rad Western Blot systems (any KD SDS-gels, Trans-Blot Turbo System) and incubated with the mentioned antibodies in Appendix Table S4 overnight at +4°C. Next, the blot was washed three times and incubated with secondary antibodies (Thermo Scientific) for 2 h and imaged with the Bio-Rad ChemiDoc Imaging System.

### Human surgical specimens

The human IRB-protocol was approved by the ethics committee of the College of Medicine at The Ohio State University. Informed consent was obtained to procure viable human surgical tissue from colon or small bowel from patients with polyps undergoing a colectomy (sigmoid colon) or patients undergoing Roux-en-Y by-pass surgery (jejunum) (Appendix Table S1). Human EGCs (hEGCs) in culture from 14 GI-surgical specimens were used to study gene expression and IL-6-release in hEGCs. Human EGCs were also used for calcium imaging studies and P2X-immunofluorescent labeling.

Collection of patient surgical specimens was also approved by the ethics committee of North-Rhine-Westphalia, Germany

(*Accession number: 266_14*). Informed consent was obtained to procure human surgical tissue from the small bowel (jejunum) from patients undergoing a pancreatectomy at an early and a late time point during the surgical procedure (Appendix Table S2). Human samples were collected and used for RNA-Seq and qPCR analysis.

### Preparation of human EGC cultures

Tissue collection was performed by the surgeon and immersed immediately in ice-cold oxygenated Krebs–Henseleit solution and promptly transported to the research facilities within 15 min in coordination with the Clinical Pathology Team (Liñán-Rico *et al*, 2016). For isolating myenteric ganglia, tissue was pinned luminal side facing upward under a stereoscopic microscope and the mucosa, submucosa and most of the circular muscle were dissected away using scissors, and then flipped over to remove longitudinal muscle by dissection.

Myenteric plexus tissue was cut and enzymatically dissociated as described elsewhere (Turco *et al*, 2014; Liñán-Rico *et al*, 2016) with modifications as follows: Myenteric plexus tissue was minced into 0.1–0.2 cm$^2$ pieces and dissociated in an enzyme solution (0.125 mg/ml Liberase, 0.5 µg/ml Amphotericin B) prepared in Dulbecco's modified Eagle's medium (DMEM)-F12, for 60 min at 37°C with agitation. Ganglia were removed from the enzymatic solution by spinning down (twice), and re-suspending in a mixture of DMEM-F12, bovine serum albumin 0.1%, and DNase 50 µg/ml DNase (once). This solution, containing the ganglia, was transferred to a 100-mm culture dish and isolated single ganglia free of smooth muscle or other tissue components were collected with a micropipette while visualized under a stereoscopic microscope and plated into wells of a 24-well culture plate and kept in DMEM-F12 (1:1) medium containing 10% fetal bovine serum (FBS) and a mixture of antibiotics (penicillin 100 U/ml, streptomycin 100 µg/mL, and amphotericin B 0.25 µg/ml) at 37°C in an atmosphere of 5% $CO_2$ and 95% humidity.

After cells reach semi-confluence after 3–4 weeks (P1), hEGCs were enriched and purified by eliminating/ separating fibroblasts, smooth muscle and other cells. EGC enrichment and purification was achieved by labeling the isolated cells with magnetic micro beads linked to anti-specific antigen, D7-Fib and passing them through a magnetic bead separation column following the manufacturer instructions (*Miltenyi Biotec Inc*, San Diego, CA). This purification protocol was performed twice (P2 and P3) to reach a cell enrichment of up to 10,000 fold, and 20,000 cells were plated on glass coverslips pre-coated with 20 µg/ml laminin/P-D-Lys in 50 mm bottom glass #0 culture dishes for immunostaining and imaging or 12 well plates for IL-6 release experiments. Cultured hEGCs were kept until confluent and harvested for additional experiments (4–10 days). On the day of the experiment, hEGCs were stimulated as indicated. Parallel to this, cells at each passage were split and seeded in plastic 25 mm$^2$ culture flasks and used for study in passages 3–6.

### Immunochemical Identification of glia in hEGC

To confirm the purity and identity of glial cells in the hEGC cultures, immunofluorescent labeling was done for glial markers (s100β, glial fibrillary acidic protein GFAP), for smooth muscle/epithelial actin and fibroblasts; hEGCs were fixed in 4% paraformaldehyde for 15 min at room temperature, rinsed three times with cold phosphate-buffered saline (PBS) 0.1 M and placed at 4°C until further processing. Cells were treated with 0.5% Triton X, 10% normal donkey serum in PBS to permeabilize the cells and block nonspecific antibody binding for 30 min at room temperature. Primary antibodies were diluted in PBS-0.1% Triton X, and 2% normal donkey serum, and were incubated with cells overnight (18–24 h) at 4°C. Next day preparations were rinsed three times in 0.1 M PBS/1 min and incubated 60 min at room temperature in secondary antibodies diluted in PBS-0.1%, Triton X, and 2% normal donkey serum. Antibodies mentioned in Appendix Table S4 were used for analysis. Alexa Fluor 488 or 568 donkey anti-mouse or anti-rabbit secondary antibodies were used at a dilution of 1:400 (Cambridge, MA). Omission of primary antibodies was used to test for background staining of the secondary antibodies. Pre-absorption of primary antisera with immunogenic peptides abolished immune-reactivity. Data confirmed previous reports by Turco *et al* (2014) and are not shown except for illustrating that all cells express s100β immunoreactivity > 99% of cells.

### *Ex vivo* human specimen experiments

Human surgical tissue for *ex vivo* experiments was collected from four patients undergoing a pancreatectomy. The study was approved by the Ethics Committee at University of Bonn. Human jejunum specimens were collected in ice-cold oxygenated Krebs–Henseleit buffer during the surgical procedure and transported to the laboratory. Full-thickness jejunum specimen were mechanically activated for 30 s and then incubated for 3 h with or without 20 µM ambroxol in oxygenated Krebs–Henseleit buffer. As baseline control a human ME sample was taken before the mechanical activation. After the incubation time, the jejunum specimens were dissected, and only mucosa-free ME was used for further analysis.

For the ethics approval, the IRB-protocol for human enteric glia isolation was approved by the ethics committee of the College of Medicine of the Ohio State University and the collection of patient material for the enteric glia analysis was approved by the ethics committee of North-Rhine-Westphalia, Germany (*Accession Number: 266_14*). Further, all experiments conformed to the principles set out in the WMA Declaration of Helsinki and the Department of Health and Human Services Belmont Report.

### Cell lines

Human 1321N1 astrocytoma cells were loaded with a calcium-chelating fluorescent dye (Molecular Devices), either fluo-4 acetoxymethyl ester (fluo-4 AM for cells transfected with P2X2, P2X4, or P2X7 receptor), or Calcium-4 AM or Calcium-5 AM for cells transfected with P2X1 or P2X3, respectively (Baqi *et al*, 2011).

HEK-P2X2 sniffer cells were loaded with 2 µM fluo-4/AM in a humidified incubator for 30 min and washed for 30 min prior to transferring to a perfusion chamber with oxygenated Krebs–Henseleit buffer on the stage of an upright Eclipse FNI Nikon scope equipped with a Andor iXon Ultra high speed camera for real-time $Ca^{2+}$ imaging. Elements software was used for data acquisition. Ambroxol was dissolved in dimethyl sulfoxide (DMSO) and added

to the cells at a final concentration of 20 μM followed by stimulation with ATP at its respective $EC_{80}$ concentration (P2X1 and P2X3 [100 nM], P2X2 and P2X4 [1 μM], P2X7 [1 mM]). The assay volume was 200 μl and the final DMSO concentration was 1%. ATP activation of the receptors led to increased calcium influx and consequently to increased fluorescence, which is blocked by treatment with antagonists.

### ELISA

Release of IL-6 and CXCL2 was measured in ME RIPA lysates isolated from small intestine segments at the indicated time points after IM. Release of IL-6 in EGC cultures incubated with various treatments was measured at the indicated time points. All ELISAs were purchased from R&D Systems (*Abingdon*, England) and used according to the manufacturer's instructions. Values were normalized to tissue weights or untreated EGCs. Briefly, for animal tissue, the isolated ME ($\sim$ 50 mg) was lysed with 1xRIPA buffer for 30 min, centrifuged for 30 min at maximum speed and the protein concentration determined with a BCA kit (Thermo Scientific). 100 μg of total protein was used to measure the release of IL-6 or CXCL2 in duplicates. For EGCs, cells were treated with the indicated substances for 24 h, supernatant was collected, centrifuged at 5,000 *g* for 5 min and snap-frozen in liquid nitrogen before processed for the IL-6 ELISA.

### RNA-Seq

RNA samples were extracted using the RNeasy Mini Kit (*Qiagen*). RNA-Seq libraries were prepared using the QuantSeq 3′ mRNA-Seq Library Prep Kit (*Lexogen*) according to the manufacturer's instructions by the Genomics Core facility of the University Hospital Bonn. The RNA samples were prepared using the QuantSeq 3′ mRNA-Seq Library Prep Kit for Illumina (*Lexogen*). The method has high strand specificity (>99.9%) and most sequences are generated from the last exon and the 3′ untranslated region. The method generates only one fragment per transcript and the number of reads mapped to a given gene is proportional to its expression. Fewer reads than in classical RNA-seq methods are needed to determine unambiguous gene expression levels, allowing a high level of multiplexing. Library preparation involved reverse transcription of RNA with oligodT primers, followed by removal of RNA and second strand cDNA synthesis with random primers. The resulting fragments containing both linker fragments were PCR amplified with primers that also contain the Illumina adaptors and sample-specific barcodes. All libraries were sequenced (single-end 50 bp) on one lane of the Illumina Hiseq 2500. Only genes with an adjusted *P*-value below 0.05 and a minimum fold change greater than 1.5 were considered to be differentially expressed between conditions.

### Immunohistochemistry

Whole mount specimens were mechanically prepared by dissection of the (sub)mucosa, fixed in 4% paraformaldehyde/PBS for 30 min, permeabilized with 0.2% Triton X-100/PBS for 15 min, blocked with 5% donkey serum/PBS for 1 h and incubated with primary IgGs mentioned in Appendix Table S4 at 4°C overnight. After three PBS washing steps, secondary antibodies (Dianova, anti-rat IgG-Cy2

1:800, anti-guinea pig IgG-Cy3, anti-chicken IgY-FITC and anti-rabbit IgG-FITC or - Cy3 1:800 were incubated for 90 min. Specimen were mounted in Fluorogel-Tris and imaged on a Leica confocal imaging system.

Primary cells were fixed in 4% paraformaldehyde/PBS for 30 min, permeabilized with 0.2% Triton X-100/PBS for 15 min, blocked with 3% BSA/BPS for 1 h and incubated with primary IgGs mentioned in Appendix Table S4 at 4°C overnight.

After three PBS washing steps, secondary antibodies (Dianova, anti-mouse IgG-Cy2 1:800, anti-guinea pig IgG-FITC and anti-rabbit IgG-FITC or - Cy3 1:800 were incubated for 60 min. Specimens were mounted in Fluorogel-Tris and imaged using a Leica confocal imaging system.

### Postoperative ileus mouse model

Postoperative ileus was induced by standardized intestinal manipulation as described previously.(Stoffels *et al*, 2014) Small bowel was eventrated after median laparotomy and gently rolled twice from oral to aboral using moist cotton swabs. After repositioning of the bowel, the laparotomy wound was closed by a two-layer suture. Two different approaches for the ambroxol administration were used, according to Migdalska-Richards *et al* (2016) animals received ambroxol (4 mM) or vehicle via drinking water starting 24 h before the surgery, until their sacrifice and according to Su *et al* (2004) animals received i.p. injections (45 mg/kg) shortly after surgery.

### *In vivo* gastrointestinal transit

Gastrointestinal transit (GIT) was assessed by measuring intestinal distribution of orally administered fluorescently labeled dextran-gavage 90 min after administration as described previously (Stoffels *et al*, 2014). The gastrointestinal tract was divided into 15 segments (stomach to colon). The geometric center (GC) of labeled dextran distribution was calculated as described previously. The stomach (st) correlates with a GC of 1, the small bowel correlates with a GC of 2–11, the cecum (c) correlates with a GC of 12 and the colon correlates with a GC of 13–15. GIT-measurement was performed with sham and IM24h animals.

### ATP measurement

ATP concentration was measured in lavage samples of naïve and POI mice at IM3h and IM24h with an ATP determination Kit (*SIGMA-Aldrich*) according to the manufacturer's instructions.

### MTT measurement

MTT signal was measured in EGCs after treating them with ambroxol, PSB1011 and PSB-0711, (all 20 μM) with a MTT assay Kit (*Abcam*) according to the manufacturer's instructions.

### MPO[+]-cell infiltration

Jejunal mucosa-free ME whole mount specimen were fixed in ethanol and stained with Hanker Yates reagent (Polyscience Europe, Germany) to identify myeloperoxidase expressing cells (MPO[+]). The mean number of MPO[+] cells/mm$^2$ for five random areas per animal

was determined. MPO$^+$ measurement was performed with naive animals, 3, 6 and 24 h after IM.

### Quantitative PCR

Total RNA was extracted from ME specimens at indicated time points after IM using the RNeasy Mini Kit (Qiagen, Hilden, Germany) followed by deoxyribonuclease I treatment (Ambion, Austin, TX). Complementary DNA was synthesized using the High Capacity cDNA Reverse Transcription Kit (Applied Biosystems, Darmstadt, Germany). Expression of mRNA was quantified by real-time RT-PCR with TaqMan probes or primers shown in Appendix Table S3.

Quantitative polymerase chain reaction was performed with SYBR Green PCR Master Mix or TaqMan Gene Expression Master Mix (both Applied Biosystems, Darmstadt, Germany).

### Fluorescence activated cell sorting

Fluorescence activated cell sorting (FACS) analysis was performed on isolated ME of the small bowel 24 h after intestinal manipulation treated with ambroxol or vehicle CX3CR1-GFP$^{+/-}$ animals, respectively. Isolation of ME was achieved by sliding small bowel segments onto a glass rod, removing the outer muscularis circumferentially with moist cotton applicators and cutting the ME into fine pieces. ME was digested with a 0.1% collagenase type II (Worthington Biochemical, Lakewood, NJ, USA) enzyme mixture, diluted in HBSS, containing 0.1 mg/ml DNase I (La Roche, Germany), 2.4 mg/ml Dispase II (La Roche, Germany), 1 mg/ml BSA (Applichem), and 0.7 mg/ml trypsin inhibitor (Applichem) for 40 min in a 37°C shaking water bath. Afterward, single cell suspension was obtained using a 70 μm filter mesh. Cells were stained for 30 min at 4°C with the appropriate antibodies. For antibodies used in this study see Appendix Table S4. Flow cytometry analyses were performed on FACSCanto III (BD Biosciences), and data were analyzed with FlowJo software (Tree Star, Ashland, OR, USA).

### Animals

Experiments were performed using 8–12-week-old mice kept in a pathogen-free animal facility with standard rodent food and tap water ad libitum. C57/BL/6J, CX3CR1-GFP$^{+/-}$ or GFAP$^{cre}$ × Ai14$^{floxed}$ mice were used for all experiments and were purchased from Jackson Laboratories directly or bred in our animal facility. All experiments were performed in accordance to the federal law for animal protection and approved by appropriate authorities of North-Rhine-Westphalia under the legal terms: 81-02.04.2018.A344 and 81-02.04.2016.A367.

### Statistical analysis

Statistical analysis was performed with Prism V6.01 (GraphPad, San Diego, CA) using Student $t$-test or one-way ANOVA as indicated. In all Figs, $P$-values are indicated as *$P < 0.05$, **$P < 0.01$ and ***$P < 0.001$ when compared to control or $^{\#}P < 0.05$, $^{\#\#}P < 0.01$ and $^{\#\#\#}P < 0.001$ when compared to indicated samples. All plots show mean + standard error of mean (SEM).

---

**The paper explained**

**Problem**

In various inflammation-induced intestinal disorders, it has been shown that reactive enteric glia play a role in disease progression by contributing to inflammatory processes. However, less is known about the underlying pathogenic mechanism of EGC activation.

**Results**

Herein, we show that enteric gliosis occurs upon abdominal surgery and leads to postoperative ileus (POI), an inflammation-based intestinal motility disorder. Activation of EGC in this process depends on ATP and selective purinergic signaling in EGCs. Within a comprehensive set of *in vivo*, *ex vivo*, and *in vitro* analyses in mice and human specimens, we found that ATP is released during abdominal surgery and activates purinergic P2X2 signaling that triggers gliosis in human and murine EGC. We further identified a novel P2X2 antagonist and P2X2 antagonism with ambroxol proved to ameliorate gliosis, reduce inflammatory responses, and improve clinical symptoms of POI.

**Impact**

We conclude that enteric gliosis and P2X-purinergic receptors might be promising drug targets for therapeutic approaches in immune-driven intestinal diseases.

Experiments were repeated with more samples when the result was close to statistical significance, and sample sizes for animal studies were chosen following previously reported studies that have used the POI animal model, at least 6–10 independent mice per experimental setup. All animals were handled by standardized housing procedures and kept in exactly the same environmental conditions and were genotyped at 4 weeks of age and received a randomized number by which they were identified. Age- and sex-matched animals were grouped randomly and used in the POI animal model. In each experimental set, all the control or experimental mice were treated with the same procedure and manipulation. For the experimental setup with ambroxol, the treatment of mice was blinded, and all mice were randomly caged. The analysis was performed on the treatment groups without knowing what group was treated with ambroxol. By this, we avoided any group or genotype-specific effects due to timing of experiments or handling of animals.

## Data availability

All RNA-seq data have been submitted to the GEO database. The datasets produced in this study are available in the following databases: RNA-Seq: mRNA from EGC and ME tissue in GSE134943 accessible here: https://www.ncbi.nlm.nih.gov/geo/query/acc.cgi?acc=GSE134943) and RNA-Seq data from mRNA of ME tissue from patients in GSE149181 accessible here: https://www.ncbi.nlm.nih.gov/geo/query/acc.cgi?acc=GSE149181; other data are available in the Expanded View Datasets or the Appendix Tables. The software tools used for this study include Partek Flow, available from https://www.partek.com/partek-flow/#features; Venn Diagramm Software, available from http://bioinformatics.psb.ugent.be/webtools/Venn/; and Gene Set

Enrichment Analysis, available from https://www.partek.com/partek-flow/#features.

Expanded View for this article is available online.

## Acknowledgements

The authors thank the Next Generation Sequencing Core Facility and the Institute for Genomic Statistics and Bioinformatics of the University Clinics Bonn for supporting the RNA-Seq analysis. The authors thank the Flow Cytometry Core Facility of the University Clinics Bonn for supporting the isolation of tdTomato[+] EGCs. The sniffer cells (HEK mixed clone 228) were a gift from Dr. Terrance Egan, Saint Louis University to Fievos L. Christofi, The Ohio State University. The gpSox10 antibody was a kind gift of Professor Wegner, University of Erlangen. NIDDKNIHR01DK113943 to Dr. Fievos L. Christofi, an NCI Cost shared resource grant P30LA16058 for the molecular core facility in the College of Medicine, The Ohio State University. This publication was supported by a personnel grant of the German research council (DFG) to SW (WE4204/3-1), BonnNI (Q-611.0754), and the ImmunoSensation[2] Cluster of Excellence (EXC 2151–390873048).

## Author contributions

RS, PL, AM, ML, IK, BS, CS, IG, EM, and EV-H performed research. RS, FLC, CEM, and SW analyzed data. RS, JCK, FLC, and SW prepared and revised the manuscript. RS, CEM, FLC, and SW designed the research. YB, CS, and CEM produced the P2X2-specific antagonists. T.G. organized the patient material.

## Conflict of interest

SW and JCK receive royalties from Wolter Kluwer for contribution to the postoperative ileus section of the *UpToDate* library. All other authors declare no competing interests.

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
