## [Review Process File · EMBO Molecular Medicine]

A novel P2X2-dependent purinergic mechanism of enteric gliosis in intestinal inflammation

Reiner Schneider, Patrick Leven, Tim Glowka, Ivan Kuzmanov, Mariola Lysson, Bianca Schneiker, Anna Miesen, Younis Bayi, Claudia Spanier, Iveta Grants, Elvio Mazzotta, Egina Villalobos-Hernandez, Jörg Kalff, Christa Müller, Fedias Christofi, and Sven Wehner

DOI: [10.15252/emmm.202012724](https://doi.org/10.15252/emmm.202012724)

Corresponding authors: Sven Wehner (Sven.Wehner@ukbonn.de)

Review Timeline:

Submission Date:	19th May 20
Editorial Decision:	29th Jun 20
Revision Received:	16th Oct 20
Editorial Decision:	5th Nov 20
Revision Received:	13th Nov 20
Accepted:	16th Nov 20

Editor: Zeljko Durdevic

Transaction Report:

29th Jun 2020

Dear Prof. Wehner,

Thank you for the submission of your manuscript to EMBO Molecular Medicine, and please accept my apologies for the delay in replying due to my recent traveling. We have now received feedback from the three reviewers who agreed to evaluate your manuscript. As you will see from the reports below, the referees acknowledge the interest of the study and are overall supporting publication of your work pending appropriate revisions.

Addressing the reviewers' concerns in full will be necessary for further considering the manuscript in our journal, and acceptance of the manuscript will entail a second round of review. EMBO Molecular Medicine encourages a single round of revision only and therefore, acceptance or rejection of the manuscript will depend on the completeness of your responses included in the next, final version of the manuscript. For this reason, and to save you from any frustrations in the end, I would strongly advise against returning an incomplete revision.

We realize that the current situation is exceptional on the account of the COVID-19/SARS-CoV-2 pandemic. Therefore, please let us know if you need more than three months to revise the manuscript.

I look forward to receiving your revised manuscript.

***** Reviewer's comments *****

Referee #1 (Remarks for Author):

It is unclear how damage reactivates enteric glial cells (EGCs) that contributes to neuroinflammation in intestine and motility disorders. This manuscript by Schneider et al. aimed to define the role of ATP and P2X2 signaling in enteric gliosis in postoperative ileus (POI). Using cell culture and intestinal manipulation (IM) models, they demonstrated that ATP-induced reactive enteric glia contribute to intestinal inflammation through P2X2. They further identified ambroxol as a novel P2X2 antagonist that abrogated enteric gliosis in a mouse IM model and human EGCs. While the study has very strong clinical relevance, significance is dampened by lack of evidence of

specificity of ambroxol and the involvement of EGCs in in vivo models.

1. Using assays in cells transfected with P2X receptors, the authors claimed that ambroxol specifically inhibits P2X2. However, if the scale of y-axis of Figure EV4 (right panel) is correct, pIC50 of ambroxol is comparable among groups. P2X2 can be knocked down/out in EGC culture to determine the effect of ambroxol is dependent on P2X2.
2. It is unclear to the review why ambroxol was screened and used in the mouse IM model, since a P2X2 antagonist (PSB-1011) is available and could be effective at a low concentration (2 μ M, Fig.2C). PSB-1011 should be used as positive control in vivo (Fig.4)
3. Ambroxol is a potent inhibitor of Na⁺/K⁺/Ca²⁺ channels and has documented anti-inflammatory effects. The authors didn't provide much evidence that the improvement of gliosis by Ambroxol was through P2X2 but not other channels.
4. IM in mice induced hundreds fold increase of CXCL2 and IL-6 in the whole ME tissue (Fig. 3G, H). However, such increase was much milder in GFAP-labelled EGCs (Fig. 3L), suggesting that other cell types might be contributing to cytokine secretion and gliosis. While the authors have determine the effect of ATP and P2X2 antagonists in culture, it is not know if the systemic treatment of Ambroxol was dependent P2X2 signaling in EGCs. EGC-specific KO of P2X2 in mice is highly desirable.
5. In Fig.1F and Fig. 3D, there were some gliosis-related genes that didn't show increase after ATP treatment or IM. What were those genes and were there any shared downregulated genes between ATP in vitro and IM in vivo?
6. Fig. 2E and 5G, P2X2 is a membrane protein but the staining didn't clearly show a membrane-enriched pattern. What did the authors do to affirm the specificity of the P2X2 antibody? Nuclear co-staining should be included in Fig. 2E.

Referee #2 (Remarks for Author):

In this study, Schneider et al., examine the role of ATP to induce enteric gliosis via the activation of P2X2 in the context of postoperative ileus and intestinal inflammation. The authors combine in vivo and in vitro experiments and include both human and mouse model systems. Although they did not identify the source of ATP, and most mechanistic data rely on the use of pharmacological agents, they make a reasonably convincing case for enteric glial ATP-P2X2 signalling after intestinal manipulation in inducing enteric gliosis, which may contribute to gut inflammation and dysmotility.

Major:

1. It is important to note that RNA sequencing data are obtained from whole mount tissue samples (muscularis externa) containing enteric neurons and immune cell infiltrates, or from cultured enteric glial cells. I suggest that the authors acknowledge this in the discussion section as an important limitation of their study.
2. I suggest the authors to validate the RNASeq data of the cultured enteric glial cells treated with ATP. For example, by the identification of genes known to be expressed in EGCs in a volcano plot or heat map, as well as negative controls (genes known to be expressed in neurons, fibroblasts,

immune cells). The PCA plot in EV1C clearly shows 2 populations of cells. Based on their molecular profile, do these 2 ATP treated clusters represent 2 different populations of enteric glial cells? How does the expression of well-known enteric glia markers differ between these two clusters? The same is actually also apparent in figure 1. Where would GFAP, NESTIN, CXCL6, IL6, FOSb and RCAN be in these plots? Same for figure 3.

3. Results section 1: "Therefore, ATP caused up-regulation in 10% and down-regulation in 11% of total glil transcriptome" where is this data?

4. The authors should be congratulated for their attempts to describe the transcriptome of the reactive enteric glia/enteric gliosis phenotype. So far, the definition of gliosis has not been well established in the enteric nervous system field but the current study is very instrumental to this end. It would be great if also some morphological characterization could be added. Is GFAP labelling (one of the hallmarks of gliosis) affected in their models?

5. The results of the current study should be discussed in relation to the findings by Boesmans et al., (2013) who showed that ATP did not change the expression of GFAP, S100 or Sox10 in a rat enteric glial cell line.

6. Cleaved caspase experiments to show cell death in the ENS have been proven difficult to interpret, and results are debated. It is not clear whether ENS cells undergo cell death via this pathway, in vivo. If the idea is to target enteric glia to prevent cell death by using the P2X2 antagonist, a simple cell viability assay (MTT) would address this matter more accurately. I would suggest this assay to be performed in the in vitro dataset using Ambroxol.

7. MHCII is not the best marker for identifying tissue resident macrophages. One reason for this is that the expression of MHCII has been detected in enteric glial cells (Geboes K et al. (1992); Koretz K et al (1987) and da Silveira AB et al. (2011)) and many others have shown MHCII in astrocytes. To be able to identify the MHCII cells as macrophages the authors will have to use another marker to identify tissue resident macrophages (e.g. CX3CR1).

8. Only few studies were able to show adult enteric gliogenesis in vivo (Laranjeira et al. (2011) and Joseph et al. (2011)). For example, after chemical ablation about 40% of enteric glia entered the cell cycle. To be able to appreciate the current results in this context, can the number of Ki67 positive enteric glia be given as a proportion of the total population? Can the authors discuss their view on the role/consequence of enteric glial proliferation in the current model? In relation to this, why did they opt for Nestin as a marker for enteric gliosis. While several studies agree on the use of Nestin as a marker of enteric glial cells, the paper by Kulkarni et al. (2017) describes Nestin as a marker for neuronal progenitors. Please comment.

9. Why was calcium imaging not performed on isolated enteric glial cells?

10. In the discussion section it is concluded that ambroxol reduced the gliosis marker increase and enteric glia proliferation by preventing leukocyte infiltration and motility impairment. Do the authors mean that the effects on enteric glia are the consequence of the reduced leukocyte infiltration and improvement of motility? Do the authors have evidence for this? Doesn't this conflict with their reasoning that ATP has direct effects on enteric glia via P2X2?

Minor:

1. Please define mEGCs. Others have used this abbreviation for mucosal enteric glia.
2. What was the timing and concentration of ATP treatment of cultured enteric glial cells?
3. All figure legends: please add which samples were used.
4. Correct typos. Introduction paragraph 5: 'later' should be latter. Results section 1, paragraph 2: 'glil' should be glial. Etc.
5. EV1F - figure legend: GFAP is violet, not red
6. Discussion: the study by Boesmans et al. (2019) shows that enteric glia respond to ATP in situ. In vitro responses were shown in Boesmans et al. (2013).

Referee #3 (Remarks for Author):

This manuscript strongly suggested that a novel pathogenic P2X2-dependent pathway of ATP-induced enteric gliosis, inflammation and dysmotility are induced at post-operative ileus in humans and mice. They concluded that block enteric glial P2X2 receptors during trauma may represent a novel therapy in treating POI and immune-driven intestinal motility disorders. A detailed analysis has been done and reasonable considerations and conclusions have been induced, but several points need to be clarified.

- 1) In whole tissue (figure 3G), the expression of CXCL2 and IL-6 peaks by IM 6h, and their expression is considerably reduced at IM 24h. On the other hand, in glial cells (figure 3 L), the expression of CXCL2 and IL-6 did not change at IM 3h, but was high at IM 24h. This difference in time course can be attributed to the inflammation of cells other than glial cells in the early stages of POI. In fact, administration of ambroxol completely suppressed the expression of IL6 at IM 24h, but the suppression at IM 3h was weak (figure 4D). In pathogeny of POI, not only neutrophils but also resident macrophages and infiltrated monocyte derived macrophages are important. One possibility is that macrophages are mainly involved in early inflammation mobilization, and then glial cell activation is also involved in late phase. Can Ambroxol reduce macrophage infiltration in POI model? You should carefully investigate movement of resident and infiltrated macrophages in POI, which in turn clarify relationship between enteric glia and macrophages in pathogeny of POI.
- 2) The increase in Sox10 positive cells expressing Ki67 is particularly greater in IM 24h than in IM 3h. Therefore, glial cell proliferation appears to be late rather than early in inflammation due to IM. On the other hand, although inflammation tends to recover in IM 24 h, what role does glial cell proliferation have in IM? Also, is the number of Sox10-positive cells actually increasing at IM 24 h?
- 3) The localization of P2X2 is difficult to understand in the in vitro immunostaining photograph in Figure 2E, so why not check not only whole-mount but also cross-section?
- 4) Activation of resident macrophages is an important origin of inflammation in POI pathology, but to what extent are glial cells involved in inflammation? How do you think about the relationship between resident macrophages? (Although it has been shown in the discussion that macrophages may be the

Schneider et al., point-by-point responses

We thank all the reviewers for their constructive comments and suggestions. We have performed additional experiments and revised the manuscript accordingly. We believe that we have addressed all the concerns raised by reviewers in the new manuscript. All changes in the revised manuscript are highlighted in yellow for your attention. Please find below our responses to the points raised by the three reviewers.

Referee #1 (Remarks for Author):

It is unclear how damage reactivates enteric glial cells (EGCs) that contributes to neuroinflammation in intestine and motility disorders. This manuscript by Schneider et al. aimed to define the role of ATP and P2X2 signaling in enteric gliosis in postoperative ileus (POI). Using cell culture and intestinal manipulation (IM) models, they demonstrated that ATP-induced reactive enteric glia contribute to intestinal inflammation through P2X2. They further identified ambroxol as a novel P2X2 antagonist that abrogated enteric gliosis in a mouse IM model and human EGCs. While the study has very strong clinical relevance, significance is dampened by lack of evidence of specificity of ambroxol and the involvement of EGCs in *in vivo* models.

1. Using assays in cells transfected with P2X receptors, the authors claimed that ambroxol specifically inhibits P2X2. However, if the scale of y-axis of Figure EV4 (right panel) is correct, pIC50 of ambroxol is comparable among groups. P2X2 can be knocked down/out in EGC culture to determine the effect of ambroxol is dependent on P2X2.

The authors appreciate the reviewer's comment about EV4 and apologize for the misunderstanding - We thank the reviewer for the good observation. The original figure EV4 was misleading, and we changed it accordingly: (see the legend in figure EV4A)

Ambroxol tested at a concentration of 20 μ M only showed significant inhibition of the P2X2, but not of the other tested P2X receptor subtypes. In order to visualize which receptors have been tested, we included all tested receptors and labeled those that have not been inhibited by ambroxol with "n.d." Higher concentrations were not studied due to the limited solubility of ambroxol and because higher plasma concentrations are typically not achieved upon treatment with ambroxol.

Furthermore, as suggested by the reviewer, we knocked down P2X2 in primary EGC cultures by a siRNA approach (see new figure EV2L+M) and could prove that glial activation by ATP is indeed dependent on P2X2-signaling (see new Figure 2F). This was specifically shown by a reduction in gene expression for GFAP, RCAN, and IL-6. For IL-6, we confirmed that protein expression measured by ELISA was reduced as well (see new Figure 2G). With regard to the specificity of ambroxol, we consider the siRNA approach not ideal, as the knockdown of P2X2 alone has a strong inhibitory effect on ATP-triggered P2X2-signaling (Figure 2E+F), and consequently, any additional effects of ambroxol are difficult to reveal.

2. It is unclear to the review why ambroxol was screened and used in the mouse IM model, since a P2X2 antagonist (PSB-1011) is available and could be effective at a low concentration (2 μ M, Fig.2C). PSB-1011 should be used as positive control *in vivo* (Fig.4).

Ambroxol was preferred because it is an established drug with a well-established bioavailability and additional pharmacokinetic properties required for *in vivo* studies (PMID: 26770490 and PMID: 14512130). In contrast, PSB-1011 is an experimental compound whose pharmacokinetic properties are unknown. PSB-1011 is suitable for *in vitro* studies but

does not possess “drug-like” properties due to its sulfonate functions, making it highly polar and may prevent cell penetration. Moreover, being an anthraquinone derivative, it can be easily reduced, yielding an inactive metabolite; in addition, it may show toxicity *in vivo*. Thus, PSB-1011 cannot be recommended, at present, for *in vivo* studies from a pharmacological perspective. We appreciate this comment, and to provide clarity on our decision, we mentioned the aspect the experimental P2X2 antagonists are not feasible for *in vivo* usage in the results part.

3. Ambroxol is a potent inhibitor of Na⁺/K⁺/Ca²⁺ channels and has documented anti-inflammatory effects. The authors didn't provide much evidence that the improvement of gliosis by Ambroxol was through P2X2 but not other channels.

The mechanism of the anti-inflammatory activity of ambroxol remains unknown. The current study provides evidence that blockade of P2X2 receptors in enteric glia is likely responsible for its anti-inflammatory activity. In keeping with this, P2X2/P2X3 antagonists have indeed been shown to be effective for the treatment of chronic cough, which is caused by chronic inflammation of the airways (doi: 10.1016/j.neuropharm.2015.12.001). In fact, the first P2X2/P2X3 antagonist, gefapixant, is in advanced clinical development for this inflammatory disorder of the airways (doi: 10.21037/jtd-20-cough-001), further strengthening our findings that ambroxol blocks inflammation by antagonism of P2X2 signaling.

Additional siRNA experiments described above prove that the ATP-induced gliosis is dependent on P2X2 in EGCs. Although we cannot totally exclude effects on other ion channels, all our data from mice and humans taken together, indicate a strong connection between ambroxol and P2X2 signaling. **This is an important point raised by the reviewer, and it is addressed in more detail in the discussion.**

4. IM in mice induced hundreds fold increase of CXCL2 and IL-6 in the whole ME tissue (Fig. 3G, H). However, such increase was much milder in GFAP-labelled EGCs (Fig. 3L), suggesting that other cell types might be contributing to cytokine secretion and gliosis. While the authors have determine the effect of ATP and P2X2 antagonists in culture, it is not know if the systemic treatment of Ambroxol was dependent P2X2 signaling in EGCs. **EGC-specific KO of P2X2 in mice is highly desirable.**

There are two reasons for the differences in the expression levels between sorted glia and whole tissue: First, flow cytometry sorting is a stressful procedure for cells, particularly after enzymatic digestion. Therefore, mRNA expression levels might be different (higher or lower) than in full tissue specimens that do not undergo this procedure. Second, there are definitely other cell types expressing IL-6 and CXCL2, e.g., macrophages and monocytes.

Regarding the specificity of ambroxol, the authors agree that an EGC-specific P2X2 knockout mouse would be the most powerful tool to prove if P2X2 is involved in EGC activation *in vivo*. Indeed, we checked all available options for these *in vivo* experiments. Unfortunately, and to the best of our knowledge, a P2X2-loxP mouse does not exist, and even pan P2X2-KO mice are not available as a living colony to check if there is a general phenotype developing in these mice in POI. Generation of a conditional P2X2 knock out mouse would need one year or more and likely, much longer with the COVID pandemic restricting our efforts, and it would include a lot of experimental expenses. This time schedule would, by far, exceed the revision time and our cost-benefit calculation, as our results

already indicate a strong connection between P2X2 and gliosis in EGCs. We already discussed this as a shortcoming of our story and included additional proof, the siRNA approach.

5. In Fig.1F and Fig. 3D, there were some gliosis-related genes that didn't show increase after ATP treatment or IM. What were those genes and were there any shared downregulated genes between ATP in vitro and IM in vivo?

Our gliosis panels in figure 1F and 3D contain both genes, which are up-regulated those who are down during glial activation. We re-analyzed the RNA-Seq data and made new Venn diagrams from up- and down-regulated genes and included them in the appendix figure 3E, and the list of these genes will be submitted as an additional dataset.

6. Fig. 2E and 5G, P2X2 is a membrane protein but the staining didn't clearly show a membrane-enriched pattern. What did the authors do to affirm the specificity of the P2X2 antibody?

For all stainings, a KO-validated P2X2 antibody was used (<https://www.alomone.com/p/anti-p2x2-receptor/APR-003>), and an IgG control staining was included (appendix figure S2F). Furthermore, we performed the staining on various samples (glial cells, intestine) with different techniques (whole-mount, paraffin), and it always showed the same reliable pattern.

In line, Guo et al., 2016 (PMID: 27105971) showed a similar staining pattern in ganglia of the small intestine and the colon.

Nuclear co-staining should be included in Fig. 2E.

As requested, we included nuclear staining in figure 2E for a better visualization of P2X2-positive EGCs and added a P2X2 staining in FFPE sections of small bowel tissue.

Referee #2 (Remarks for Author):

In this study, Schneider et al., examine the role of ATP to induce enteric gliosis via the activation of P2X2 in the context of postoperative ileus and intestinal inflammation. The authors combine in vivo and in vitro experiments and include both human and mouse model systems. Although they did not identify the source of ATP, and most mechanistic data rely on the use of pharmacological agents, they make a reasonably convincing case for enteric glial ATP-P2X2 signalling after intestinal manipulation in inducing enteric gliosis, which may contribute to gut inflammation and dysmotility.

Major:

1. It is important to note that RNA sequencing data are obtained from whole mount tissue samples (muscularis externa) containing enteric neurons and immune cell infiltrates, or from cultured enteric glial cells. I suggest that the authors acknowledge this in the discussion section as an important limitation of their study.

We appreciate this suggestion and revised the discussion section to clarify the limitations of using bulk RNA-seq. in our study.

2. I suggest the authors to validate the RNASeq data of the cultured enteric glial cells treated with ATP. For example, by the identification of genes known to be expressed in EGCs in a volcano

plot or heat map, as well as negative controls (genes known to be expressed in neurons, fibroblasts, immune cells).

We guessed that the reviewers' question aims for an additional check of the purity in our primary EGC cultures. To address this point, we generated a new heat map with specific markers for glia, neurons, fibroblasts/epithelial, and immune cells and included it as appendix figure S1A. However, with the immunocytochemistry in figure EV1A, we already showed that our culture is not 100% pure and that it includes around 15% α -SMA-positive, non-glia cells. Accordingly, the new heat map shows that some non-glia markers are expressed, but the glial markers are still enriched in our cultures. Sadly, the purity is always one limitation of primary cell cultures, and we are aware of this problematic. Notably, to make the reviewer aware of the quality of our cultures, we put this information at the beginning of the results part.

The PCA plot in EV1C clearly shows 2 populations of cells. Based on their molecular profile, do these 2 ATP treated clusters represent 2 different populations of enteric glial cells? How does the expression of well-known enteric glia markers differ between these two clusters?

We thank the reviewer for these questions, and we analyzed the two ATP-treated populations again to look for the expression of glial markers and create a new heat map. The expression of glial markers seemed to be equally expressed in both groups.

Next, we re-analyzed our *in vitro* data and added a 3rd dimension in our PCA (Fig EV1C). The new analysis still showed a difference in both ATP-treated groups and a homogeneous control group. By re-evaluating the experimental scheme, we concluded that the two ATP-treated groups came from two different cell batches. Whereas the controls from these two batches showed no strong differences, the ATP-treatment seems to induce a separation. Although working with a standardized protocol and procedure, primary cell cultures always contain a certain degree of contaminating non-glia cells that can change the PCA. In our opinion, the discrepancy between the two groups can occur due to the effects of the non-glia cells in the culture and through a stronger or weaker activation of our glia cells by ATP. We do not think that our culture conditions produce different populations of glial cells. Obviously, our *in vitro* system was stable enough for the expression of standard glial markers and our "core" marker set of gliosis genes; only the whole transcriptome showed differences in the EGC cultures. To overcome exactly this limitation, we further tested our *in vitro* hypothesis in an *in vivo* POI model.

The same is actually also apparent in figure 1. Where would GFAP, NESTIN, CXCL2, IL6, FOSb and RCAN be in these plots? Same for figure 3.

We created two new heat maps for the RNA-Seq data of EGCs *in vitro* and *in vivo* showing the mentioned genes. To visualize these genes in the main figures seemed to be overly complicated, so we included them in the **appendix figures S1C and S3F**.

3. Results section 1: "Therefore, ATP caused up-regulation in 10% and down-regulation in 11% of total glial transcriptome" where is this data?

The mentioned percentages are concluded from the relation between total genes (around 20000) divided by up (2027) and down (2218) regulated genes. These data are presented in the volcano plot in figure 1.

4. The authors should be congratulated for their attempts to describe the transcriptome of the reactive enteric glia/enteric gliosis phenotype. So far, the definition of gliosis has not been well established in the enteric nervous system field but the current study is very instrumental to this end. It would be great if also some morphological characterization could be added. Is GFAP labelling (one of the hallmarks of gliosis) affected in their models?

The authors thank the reviewer for agreeing on the findings of this important effort. To address further this matter, we investigated morphological changes by analyzing GFAP histology *in vivo*. Accordingly, we investigated any morphological changes in our animal model at IM24h, as glial activation peaks at this time point. By comparing GFAP-stained ganglia from naïve and IM24h animals, we were able to analyze ganglia structures and the impact of glial cell morphology. Our histology showed a distinct difference at IM24h with a stronger GFAP signal and more branching in the ganglia (we included this as figure EV3E), but it was challenging to develop a reliable quantification method for this *in vivo* analysis. Morphological analysis is a common tool to define the activity of various cell types like immune and neural cells, and the used readouts became more complex in the last years (PMID: 27226303). In the gut, these tools are not established, and it is far from simple to transfer methodical approaches developed in the CNS (PMID: 30929313) to the ENS. In our opinion, this topic deserves an extensive study on its own and goes far beyond what we can address in this revision.

5. The results of the current study should be discussed in relation to the findings by Boesmans et al., (2013) who showed that ATP did not change the expression of GFAP, S100 or Sox10 in a rat enteric glial cell line.

The authors looked into the study of Boesmans et al. and discovered some essential differences between our two studies. In Boesmans et al., the treatment scheme was 50µM ATP for 24 hours; we could show that strong gliosis effects are seen with 100µM ATP after 6 hours. Therefore, we decided to subject the cell line used by Boesman et al. to our treatment scheme:

Our results show a significant but rather weak upregulation of GFAP and IL-6 after 100µM ATP treatment for 6h. By comparing the P2X2 expression by qPCR, we discovered that the P2X2 expression is far lower in the rat glial cell line than in primary mouse glia. Consequently, we think that this rat cell line is not suitable as an *in vitro* model to evaluate ATP-induced gliosis via P2X2 receptors. The difference could involve both species differences and a difference between primary cells in culture versus a cell line.

6. Cleaved caspase experiments to show cell death in the ENS have been proven difficult to interpret, and results are debated. It is not clear whether ENS cells undergo cell death via this pathway, *in vivo*. If the idea were to target enteric glia to prevent cell death by using the P2X2 antagonist, a simple cell viability assay (MTT) would address this matter more accurately. I would suggest this assay to be performed in the *in vitro* dataset using Ambroxol.

The authors agree that supportive data are required and performed an MTT assay in primary cultures treated with all P2X2-antagonists used in this publication. The MTT results are in line with the immunohistochemistry data showing absence of cleaved caspase-3 and showed no toxic effect of any P2X2 antagonist. (appendix figure 2E+4E).

7. MHCII is not the best marker for identifying tissue resident macrophages. One reason for this is that the expression of MHCII has been detected in enteric glial cells (Geboes K et al. (1992); Koretz K et al (1987) and da Silveira AB et al. (2011)) and many others have shown MHCII in astrocytes. To be able to identify the MHCII cells as macrophages the authors will have to use another marker to identify tissue resident macrophages (e.g. CX3CR1).

Although MHCII clearly identifies the resident muscularis macrophages without any notable staining of other cell types in our hands, and MHCII was also used for the same purpose very recently by Grubisic et al., 2020 (PMID:32905782), the authors agree that CX3CR1 is a more specific macrophage marker than MHCII. Therefore, we subjected CX3CR1-GFP mice to the POI model and performed new immunohistological stainings for naïve, IM3h, IM6h, and IM24h timepoints (appendix figure 3D). The staining shows CX3CR1-positive macrophages surrounding enteric glial cells in naïve animals and in the disease course.

8. Only few studies were able to show adult enteric gliogenesis *in vivo* (Laranjeira et al. (2011) and Joseph et al. (2011)). For example, after chemical ablation about 40% of enteric glia entered the cell cycle. To be able to appreciate the current results in this context, can the number of Ki67 positive enteric glia be given as a proportion of the total population?

The authors re-calculated all Sox10 stainings and normalized the proliferating (Ki67-positive) enteric glia to the total number of Sox10 cells in the ganglia (Figure 3I + 4E).

9. Can the authors discuss their view on the role/consequence of enteric glial proliferation in the current model?

In the brain, the increased proliferation in reactive astrocytes leads to the typical phenotype of glial scarring in the CNS. For the enteric nervous system, not much is known about glial proliferation under pathological conditions; mainly basal conditions are discussed and show almost no glial proliferation in healthy animals. We can only speculate that under inflammation, this induced proliferation is important in recovery processes. In this way, enteric glia could contribute to the healing of the damaged ENS activity. In POI, mice recover rapidly, mainly after 48-72h after the induction of the motility impairment. Therefore, we focused only on early (IM3h-6h) and acute phases (IM24h) but not on the remission phase. We were rather interested in the “reactivity” itself and not in the long-term function. Future studies in chronic and long-lasting disease models, i.e., in IBD models, should focus on the recovery phase to better assess the impact of enteric glial proliferation. However, we believe that this is an important aspect, and therefore we discussed it in more detail in the revised manuscript.

In relation to this, why did they opt for Nestin as a marker for enteric gliosis. While several studies agree on the use of Nestin as a marker of enteric glial cells, the paper by Kulkarni et al. (2017) describes Nestin as a marker for neuronal progenitors. Please comment.

The authors chose Nestin as a marker for gliosis because it is already used in the CNS as a prominent marker for reactive astrocytes. Kulkarni et al. (2017) described Nestin as a marker for neuronal progenitors, but their *in vivo* analysis also revealed that the Nestin-positive cells also express S100b, a typical glial marker, making these cells rather complicated to define. Induction of Nestin in enteric glia could indicate a de-differentiation towards a glial progenitor, or the protein is a simple marker of stress in these cells. At the beginning of our study, we tested many different gliosis markers known from CNS literature, and we discovered that Nestin is a reliable marker for ATP-induced enteric gliosis.

10. Why was calcium imaging not performed on isolated enteric glial cells?

Thank you for this question. We are very much interested in conducting calcium imaging experiments in isolated enteric glial cells (mice and humans), as well as calcium reporter mice *in situ*, to further evaluate the role of glial P2X2 receptors in postoperative ileus, using

the calcium wave as the physiological readout for the response. It is well known that manipulating calcium waves in glia influences motility, and studies by Gulbransen's group have shown this in multiple studies over the past decade. However, as would be expected, these experiments are quite challenging to do.

We have given this a lot of thought, and it is in our future plans, but there are a number of hurdles to overcome to study P2X2 receptors. First, it is clear that there are multiple purinergic receptors on enteric glia in mice and humans, and it is exceedingly difficult to use the calcium wave to study the role of P2X2 in ATP responses and its blockade with ambroxol, or other antagonists in mice or humans. It is proving to be an endpoint in itself. Calcium waves is a general physiological readout, and it would not be simple or straight forward to tease out a P2X2 component in this way in normal or inflamed gut (unlike targeting IL6, for example, a proinflammatory endpoint, and which may represent a more robust readout for P2X activation in the inflamed state). Certainly, such studies on calcium waves would ultimately, at the very least, provide complementary data linking alterations in purinergic calcium waves mediated by P2X2 receptors in glia to gliotransmission and dysmotility in POI. Whether such actions on P2X2 signals are related in a direct way to enteric gliosis is another story, and for that reason, may not necessarily provide additional direct evidence in support of our hypothesis on enteric gliosis.

Furthermore, from a practical standpoint, studies on calcium waves in mouse and human enteric glia in normal and inflamed states to test the role of P2X2 receptors would require a detailed, comprehensive pharmacological analysis in control, sham and postoperative ileus mice at different time points during the development of POI, or control and inflamed hEGCs (Linan-Rico et al., Christofi, 2016, IBD) or intact human gut preparations for P2X receptors (Linan-Rico et al. - Christofi, Neuropharmacology, 2015). It is also possible that glial P2X receptors are up-regulated in their expression in postoperative ileus at different time points during the development of the disease after gut surgical manipulation. Therefore, we think that to precisely address this important question, it would be best to do a separate comprehensive analysis that stands alone and is beyond the scope of the current study, that is already fairly robust with cellular targets, approaches, and analyses, and provides multiple lines of evidence for glial P2X2 signaling in enteric gliosis and POI. It would also require the use of siRNA's and development of glial specific P2X2- knockout mice, which do not exist as yet, to unequivocally confirm the involvement of the P2X2 cellular target in abnormal calcium waves and the pathogenesis of POI. A selective P2X2 receptor agonist does not exist for these studies, and it would be extremely useful. Finally, we restricted calcium experiments to human P2X2-sniffer cells and other cells expressing P2X2 to evaluate the antagonistic potency of ambroxol and other P2X2 antagonists at these receptors.

11. In the discussion section it is concluded that ambroxol reduced the gliosis marker increase and enteric glia proliferation **by** preventing leukocyte infiltration and motility impairment. Do the authors mean that the effects on enteric glia are the consequence of the reduced leukocyte infiltration and improvement of motility? Do the authors have evidence for this?

The authors thank the reviewer for the chance to clarify this key point of our study. In our understanding, activated glial cells are contributing to leukocyte infiltration and motility impairment, and the dampening of gliosis by ambroxol results in less leukocyte infiltration and improved motility. To clarify, we changed the word "by" to the word "and". To this matter, Grubisic et al., 2020 (PMID:32905782) recently showed that enteric glia modulate the macrophage phenotype in chronic inflammation, and Rao et al., 2017 (PMID:28711628)

already described the contribution of EGCs to gut motility. Therefore, both studies are in line with our publication. Consequently, we included this in the revised discussion part and cited the additional supporting literature.

Doesn't this conflict with their reasoning that ATP has direct effects on enteric glia via P2X2?

No, our mechanism is built on the activation of EGCs through P2X2 by ATP. This activation leads to gliosis and transforms enteric glia into "reactive glia" that contribute to the impaired motility and other inflammatory mechanisms in the gut.

Minor:

1. Please define mEGCs. Others have used this abbreviation for mucosal enteric glia.

In our study, mEGCs are murine EGCs isolated from the muscularis externa of the small intestine. To make it simpler, we changed our abbreviation from mEGC to msEGC, as we mainly needed a clear distinction between human and mouse EGCs.

2. What was the timing and concentration of ATP treatment of cultured enteric glial cells?

As mentioned in the figure legend: 6h for transcriptional analysis or 24h for ELISA with 10 or 100 μ M ATP or ATP γ S.

3. All figure legends: please add which samples were used.

We changed all figure legend accordingly.

4. Correct typos. Introduction paragraph 5: 'later' should be latter. Results section 1, paragraph 2: 'glil' should be glial. Etc.

Ok.

5. EV1F - figure legend: GFAP is violet, not red

We apologize for the mistake and corrected the figure legend.

6. Discussion: the study by Boesmans et al. (2019) shows that enteric glia respond to ATP in situ. In vitro responses were shown in Boesmans et al. (2013).

We apologize for this mistake and corrected the citations.

Referee #3 (Remarks for Author):

This manuscript strongly suggested that a novel pathogenic P2X2-dependent pathway of ATP-induced enteric gliosis, inflammation and dysmotility are induced at post-operative ileus in humans and mice. They concluded that block enteric glial P2X2 receptors during trauma may

represent a novel therapy in treating POI and immune-driven intestinal motility disorders. A detailed analysis has been done and reasonable considerations and conclusions have been induced, but several points need to be clarified.

1) In whole tissue (figure 3G), the expression of CXCL2 and IL-6 peaks by IM 6h, and their expression is considerably reduced at IM 24h. On the other hand, in glial cells (figure 3 L), the expression of CXCL2 and IL-6 did not change at IM 3h, but was high at IM 24h. This difference in time course can be attributed to the inflammation of cells other than glial cells in the early stages of POI. In fact, administration of ambroxol completely suppressed the expression of IL6 at IM 24h, but the suppression at IM 3h was weak (figure 4D). In pathogeny of POI, not only neutrophils but also resident macrophages and infiltrated monocyte derived macrophages are important. One possibility is that macrophages are mainly involved in early inflammation mobilization, and then glial cell activation is also involved in late phase.

1. Can Ambroxol reduce macrophage infiltration in POI model?

We thank the reviewer for these comments. To address this question, we used the CX3CR1-GFP transgenic mouse line in our POI model. These mice allow us to distinguish macrophages from monocytes (PMID: 29291892) and help us to understand any impact of ambroxol on the infiltrate during POI. In the POI model, the infiltration is highest at IM24h, so we analyzed this time point within our treatment scheme with ambroxol (Figure 4A). In general, our FACS analysis showed a reduction of both monocytes and macrophages in POI after ambroxol treatment, an additional confirmation for the MPO-histology on muscularis externa specimens from these mice. We will include this data together with the gating strategy in figure 4G and EV4D, respectively. These new results indicate that ambroxol can modulate the infiltrate and reduces the total amount of monocytes and macrophages. These data serve as a fitting addition to our hypothesis that a dampened gliosis affects immune cell infiltration in POI.

2. You should carefully investigate movement of resident and infiltrated macrophages in POI, which in turn clarify relationship between enteric glia and macrophages in pathogeny of POI.

The movement or migration of macrophages within solid tissue in an *in vivo* model is not easy to access. Our FACS data from CX3CR1 mice in POI showed already that the amount of immune cells is reduced after ambroxol treatment. To visualize the location of CX3CR1 cells towards enteric glial cells, we also performed confocal images at IM24h. By this, we could access at least the amount of immune cells surrounding ganglia. In line with the FACS quantification, we discovered less CX3CR1 cells in proximity of enteric glia after ambroxol treatment in POI. We will include the histology of CX3XR1-GFP and GFAP as figure EV4E.

3. The increase in Sox10 positive cells expressing Ki67 is particularly greater in IM 24h than in IM 3h. Therefore, glial cell proliferation appears to be late rather than early in inflammation due to IM.

On the other hand, although inflammation tends to recover in IM 24 h, what role does glial cell proliferation have in IM?

See also reviewer 2 Q9

In the brain, the increased proliferation in reactive astrocytes leads to the typical phenotype of glial scarring in the CNS. For the enteric nervous system, not much is known about glial proliferation under pathological conditions; mainly basal conditions are discussed and show almost no glial proliferation in healthy animals. We can only speculate that under inflammation, this induced proliferation is important for processes important in recovery processes. In this way, enteric glia could contribute to the healing of the damaged ENS activity. In POI, mice recover rapidly, mainly after 48-72h after the induction of the motility impairment. Therefore, we focused only on early (IM3h-6h) and acute phases (IM24h) but not on the remission phase. We were rather interested in the "reactivity" itself and not in the long-term function. Future studies in chronic and long-lasting disease models, i.e., in IBD models, should focus on the recovery phase to better access the impact of enteric glial proliferation. However, we believe that this is an important aspect, and therefore we discussed it in more detail in the revised manuscript.

4. Also, is the number of Sox10-positive cells actually increasing at IM 24 h?

See also Reviewer 2 Q8.

The number of Sox10 cells is not increasing during the disease course; only the amount of proliferating cells is significantly increased. We re-calculated all Sox10 stainings and normalized the proliferating (Ki67-positive) EGC to the total number of Sox10⁺ cells (**Figure 3I**). In our opinion, 24h are not enough time to increase the number of glia in the intestine; a 72-96h time point could answer this question far better.

5. The localization of P2X2 is difficult to understand in the in vitro immunostaining photograph in Figure 2E, so why not check not only whole-mount but also cross-section?

See also Reviewer 1 with Q6.

For an additional, better visualization of P2X2-positive EGCs, we made an additional P2X2 staining in FFPE sections of small bowel tissue (Addition for Figure 2E).

6. Activation of resident macrophages is an important origin of inflammation in POI pathology, but to what extent are glial cells involved in inflammation? How do you think about the relationship between resident macrophages? (Although it has been shown in the discussion that macrophages may be the ATP source).

The authors agree with the reviewer that the connection between macrophages and enteric glia is indeed an interesting aspect. However, we feel that is also a topic on its own, and appropriate experiments clarifying the role and significance of this relationship is beyond the scope of our manuscript. Nevertheless, we re-analyzed samples previously published by our group (Stein et al., 2018, PMID: 30581430) from CCR2-KO mice used in the POI mouse model. These mice suffer from the same motility impairment at IM24h but miss monocyte infiltration, which are the most prominent infiltrating immune cells in POI. By analyzing enteric gliosis in these mice, we could gain some insight into the crosstalk between resident macrophages and glia after the mechanical activation by excluding any effects generated by infiltrating cells (PMID: 28615302). In general, gliosis markers are dampened in CCR2-KO mice during POI compared to WT mice (see new Appendix Figure S4F), showing that leukocyte infiltration and the following inflammatory stimulus increases glial activation in the disease progress and "keep the gliosis going". These results also show that the initial glial

activation is independent of the infiltrate and happens shortly after or in parallel to the resident macrophage activation. Macrophages are in close contact with EGCs, and their numbers increase during POI (see new Appendix Figure S3D), and both cell types seem to be activated at the beginning of POI and work together during the local inflammation. The motility impairment, however, is not connected to the infiltrate, and even a dampened gliosis in the CCR2-KO mice was sufficient to induce it. Consequently, initial activation by, e.g., mechanical stimuli is enough to disturb the gastrointestinal homeostasis (macrophage/glia relationship) to impair ENS function (PMID: 28615301 and PMID: 28615302). We included these data in appendix figure 4F and addressed them in our discussion section. While these data do not fully clarify the role of macrophage and glial cell interaction, they at least show that gliosis is affected by blood-derived monocytes and macrophages but also occurs when only resident macrophages are present. Further studies on the role of resident macrophage and enteric glial cell interaction are warranted.

5th Nov 2020

Dear Prof. Wehner,

Thank you for the submission of your revised manuscript to EMBO Molecular Medicine. I am pleased to inform you that we will be able to accept your manuscript pending the following final amendments:

***** Reviewer's comments *****

Referee #1 (Remarks for Author):

The revised manuscript answered all my questions.

Referee #2 (Comments on Novelty/Model System for Author):

NA

Referee #2 (Remarks for Author):

I would like to thank the authors for their efforts in addressing my concerns and questions, and congratulate them for an important and compelling study. I have no further comments.

The authors performed the requested changes.

The authors performed the requested changes.

Corresponding Author Name: Prof. Sven Wehner

Manuscript Number: EMM-2020-12724